# MATCHED DATA, BETTER MODELS: TARGET ALIGNED DATA FILTERING WITH SPARSE AUTOENCODERS

**Arnav M. Das**,* **Gantavya Bhatt**,* **Sahil Verma,** **Yiping Wang**
**Viswa Virinchi Muppirala**, **Jeff Bilmes**

University of Washington

## ABSTRACT

Data filtering plays a central role in improving model performance, particularly for vision language models that are pretrained on large, noisy, and redundant image-caption datasets. Existing filtering techniques assess every sample individually and retain those that exceed a certain quality threshold, but such strategies fail to capture higher-order interactions. In this work, we propose a novel submodular framework for data selection that addresses this limitation. Our method, Submodular Distribution Matching (SDM), selects a subset by: (1) training a type of sparse autoencoder to learn disentangled and *monotone* features; (2) estimating a target feature distribution from a target dataset; and (3) selecting a subset of samples whose feature distribution closely matches the target via submodular maximization. Given the DataComp-medium training set and no external models, SDM achieves state-of-the-art accuracy on both ImageNet-1K and average performance across 38 downstream tasks. On the full DataComp-medium benchmark, SDM delivers performance within 1% of the state-of-the-art results while using over **5×** fewer GPU hours than the leading approach.

## 1 INTRODUCTION

Web-scale image-caption datasets have been critical to recent advances in multimodal learning, enabling capabilities such as zero-shot image classification (Jia et al., 2021; Radford et al., 2021), text-guided generation (Kim et al., 2022; Ramesh et al., 2021; Zhang et al., 2023), multimodal retrieval (Radford et al., 2021), and a range of other applications (Jiang et al., 2023). However, because these datasets are typically scraped from the web, they contain noisy and redundant samples that can degrade model performance (Elazar et al., 2024; Webster et al., 2023). Given their scale, manual curation is infeasible, making algorithmic data selection an increasingly important area of research.

Most data selection methods applied at web-scale follow a common strategy: estimate the quality of each sample and retain those above a threshold. Some methods use simple heuristics, such as image resolution or caption length (Gadre et al., 2023); others leverage CLIP embeddings to assess semantic alignment (Gadre et al., 2023; Wang et al., 2024b) or proximity to a reference dataset like ImageNet (Gadre et al., 2023; Wang et al., 2024b). More recent approaches train specialized models on curated external datasets to estimate sample quality (Fang et al., 2024; Kim et al., 2024; Shechter & Carmon, 2025b). In general, these methods treat each sample independently when evaluating their utility.

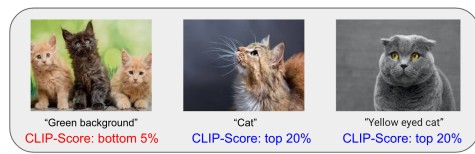

Figure 1: **Quality Score Limitations**. Low-quality samples can still be useful for learning broader concepts. For example, a model may have enough samples to learn the concept of "cat," but without the low-scoring "green background" image, it lacks examples needed to learn the concept of "green."

While evaluating samples independently can be computationally efficient, it overlooks properties that only emerge at the dataset level. For example, individually high-quality samples may be redundant

---

*Equal contribution. Correspondence to {arnavmd2,gbhatt2}@uw.edu.

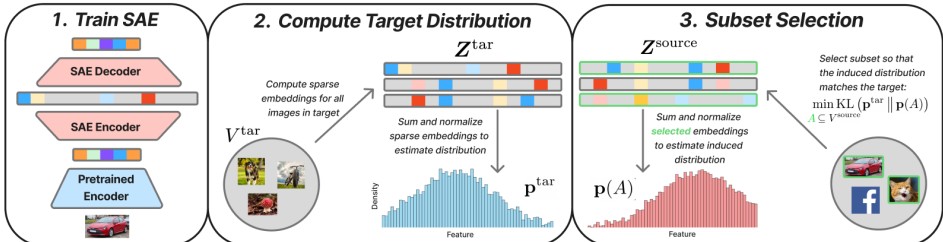

Figure 2: **Overview**. Our pipeline consists of three steps: **(1)** Train a sparse autoencoder (SAE) to disentangle pretrained neural features into meaningful sparse representations, **(2)** estimate an empirical distribution of a target dataset over sparse features **(3)** use a submodular distribution matching objective to select $A \subseteq V$ that matches the target distribution.

with each other when selected together, while samples that seem low-quality in isolation may capture rare concepts that enhance diversity when included (see Figure 1). Ignoring distributional properties during selection can lead to datasets that are imbalanced, ultimately limiting the generalization ability of models trained on them (Aghabagherloo et al., 2025).

*We argue that effective data selection must also account for distributional properties, particularly by ensuring a balanced representation of underlying concepts at the set level.* This goal requires us to (1) identify and reason about the presence of individual concepts at a set level and (2) select a subset of data such that the distribution over these concepts exhibits desirable properties.

Standard neural representations, such as CLIP embeddings, are ill-suited for this purpose. These representations tend to be entangled (Olah et al., 2020), blending multiple concepts in ways that make it difficult to quantify or control how much of a given concept is present in a set. To address this, we use sparse autoencoders (SAEs) to disentangle compact neural representations into sparse features that correspond to disentangled concepts (Makhzani & Frey, 2014). We further require these features to be *monotone*, meaning that their values increase as more of the corresponding concept is present (Gupta et al., 2016). This encourages the feature values behave more like (fractional) counts: when aggregating over a set, the total contribution of a concept can be obtained simply by summing its values across individual items. To this end, we introduce a novel loss term inspired by the "peripteral" loss introduced in Bhatt et al. (2024) to promote learning monotone features. Intuitively, these disentangled and monotone features provide a set of "knobs" we can adjust to control which concepts—and in what proportions—are included in the final dataset.

To balance concepts in a desirable way, we frame data selection as a distribution matching problem: our goal is to match the distribution over concepts in the selected set to a target distribution derived from a downstream task. We then connect this distribution matching objective to submodular maximization: specifically, we show that maximizing certain instances of feature-based functions (FB functions)—a class of monotone, non-decreasing submodular functions—correspond to minimizing the KL divergence between the concept distribution of the selected set and the target. This connection allows us to use scalable algorithms for approximate maximization with constant factor guarantees (Nemhauser et al., 1978). Finally, we demonstrate that this objective can be combined with existing quality-based filtering strategies, enabling a unified approach that considers both distributional properties and individual sample quality.

**Our contributions can be summarized as follows:** (1) We introduce a data selection framework that disentangles dense features into disentangled and monotone concepts and selects a subset of data that maxmimizes a FB function based objective. (2) We offer a theoretical interpretation of the FB objective by connecting it to distribution matching. (3) We demonstrate strong empirical results on DataComp-medium, a dataset of 128M image-caption pairs, showing that models trained on our filtered dataset achieve significant improvements compared to existing data selection methods.

## 2 METHOD

In this section, we begin with a brief introduction to feature-based functions (FB functions) (§2.1), describe the process we use to construct a disentangled and monotone set of concepts (§2.2 §2.3), and finally derive the proposed subset selection objective (§2.4, §2.5). Our full workflow is shown in Figure 2. We also provide a brief introduction to submodularity and a scalable strategy commonly used to approximately maximize these functions in Appendix C.

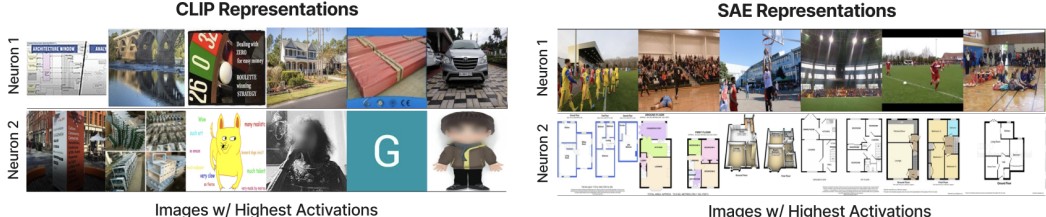

Figure 3: **SAE Visualization.** Each row displays the top-activating images from DataComp for two randomly selected neurons of the CLIP image encoder (left) and an SAE (right). CLIP features tend to activate for a mix of unrelated concepts, while SAE features represent more coherent concepts. See Appendix L for more examples.

**Notations.** Unless mentioned otherwise, $V = V^{\text{source}} = [n] = \{1, 2, \dots n\}$ denotes the ground set indexing the data points to select a subset from, and $V^{\text{tar}}$ denotes the ground set indexing the target data points. We denote the space of latent representation of images as $\mathcal{X} \subseteq \mathbb{R}_+^{d_{\text{in}}}$ and the space of representation of sparse representations output as $\mathcal{Z} \subseteq \mathbb{R}_+^{d_{\text{sparse}}}$. $\boldsymbol{Z}^{\text{source}}$ and $\boldsymbol{Z}^{\text{tar}}$ are the design matrices for source and target respectively, such that $\boldsymbol{Z}_j^{\text{source}} \triangleq h^{\text{enc}}(\boldsymbol{x}_j^{\text{source}})^T$ (similarly for the target), where $h^{\text{enc}} : \mathcal{X} \to \mathcal{Z}$ denotes the encoder of sparse autoencoder. $\Delta^{d-1}$ represents a probability simplex in $d$-dimensions and $\mathbb{I}[\cdot]$ is an indicator function. Lastly, we use $\phi$ to denote the concave function, and $m(A)$ represents a modular function defined on any $A \subseteq V$.

## 2.1 PRIMER ON FEATURE-BASED FUNCTIONS

This work focuses on a specific class of submodular function referred to as *feature-based functions* (Stobbe & Krause, 2010; Bilmes & Bai, 2017) or FB functions. Given a set of indices $V = [1, \dots, n]$, a design matrix $\boldsymbol{Z} = [\boldsymbol{z}_1, \dots, \boldsymbol{z}_n]^\top \in \mathbb{R}_+^{n \times d}$ of non-negative entries, a list of monotone, non-decreasing concave functions $(\phi_i)_{i=1}^n$, and a vector of non-negative weights $\boldsymbol{w} \in \mathbb{R}_+^d$ we define feature-based functions for any $A \subseteq V$ as follows:

$$f(A) = \sum_{i=1}^{d} w_i \, \phi_i\big(m_i(A)\big), \quad \text{where} \quad m_i(A) \triangleq \sum_{j \in A} z_{ji}, \tag{1}$$

For FB-functions, each feature must be *monotone*, meaning that larger $z_{ji}$ values correspond to a stronger presence of concept $i$ in item $j$, and larger $m_i(A)$ indicates greater total presence of concept $i$ in set $A$. For instance, in bag-of-words or TF-IDF-like representations, $z_{ji}$ is the frequency (or weighted frequency) of word $i$ in document $j$, so $m_i(A)$ naturally increases as more documents in $A$ contain that word. The concavity of each $\phi_i$ enforces diminishing returns: once a concept is already well represented in set $A$, adding more instances with similar concepts contributes less marginal gain. Finally, the weights $w_i$ allow assigning relative importance to different features.

## 2.2 OBTAINING SPARSE FEATURES

We begin by extracting information-rich representations from a pretrained image encoder; however, these features typically respond to several, potentially unrelated concepts (Olah et al., 2020). To address this, we disentangle these representations using Sparse Autoencoders (SAEs), drawing inspiration from early sparse coding methods (Lee et al., 2006) and recent advances in interpretability (Makhzani & Frey, 2014; Gao et al., 2025; Huben et al., 2024; Pach et al., 2025).

Formally, given an input $x$ already in some dense representation space $\mathcal{X} \subseteq \mathbb{R}_+^{d_{\text{in}}}$, we define a $k$-SAE (Gao et al., 2025; Makhzani & Frey, 2014) with autoencoding in $d_{\text{sparse}}$ dimensions using an encoder $h^{\text{enc}}$ and an affine mapping based decoder, respectively, with encoder projecting input into a sparse, high dimensional non-negative space ($d_{\text{sparse}} >> d_{\text{in}}$).

$$h^{\text{enc}}(x) = \text{TopK}\big(\text{ReLU}(W_{\text{enc}}(x - b_{\text{dec}}) + b_{\text{enc}})\big); \quad \tilde{x} = W_{\text{dec}} \, h^{\text{enc}}(x) + b_{\text{dec}} \tag{2}$$

where $W_{\text{enc}} \in \mathbb{R}^{d_{\text{sparse}} \times d_{\text{in}}}$, $b_{\text{enc}} \in \mathbb{R}^{d_{\text{sparse}}}$, $W_{\text{dec}} \in \mathbb{R}^{d_{\text{in}} \times d_{\text{sparse}}}$, $b_{\text{dec}} \in \mathbb{R}^{d_{\text{in}}}$ and $\text{TopK}(\cdot)$ only retaining top-$k$ entries of a vector and setting anything else as 0. In other words, every input vector is transformed by the encoder into a sparse representation that has $k$ nonnegative entries

and reconstructs $\tilde{x}$, using an affine transformation. $k$-SAE's are typically trained with a simple reconstruction loss $\mathcal{L}_{\text{recons}}(x, \tilde{x}) = \|x - \tilde{x}\|_2^2$ and we include an additional activity regularizer $\mathcal{L}_{\text{act-reg}} = \sum_{x \in b} \|h_\phi(x)\|^2 / |b|$ to keep activations from being too large (Merity et al., 2017). The resulting sparse representations tend to exhibit the property that individual features respond to images containing a concept which can often be interpretable by humans as shown in Figure 3 - this property is frequently referred to as *monosemanticity* (Gao et al., 2025; Huben et al., 2024; Pach et al., 2025).

We note that the property of monosemanticity is ill-defined, since it depends entirely on how one chooses to define a "concept." For instance, a hidden neuron that activates in the presence of any bird in an input image can be considered monosemantic with respect to the bird concept, but polysemantic with respect to individual bird species. Nevertheless, we use the shorthand "monosemanticity" to mean so with respect to a fixed set of concepts.

Monosemanticity is also distinct from the notion of monotonicity. For example, a hidden neuron that only activates in the presence of a bird may still be considered monosemantic even if its activation value is higher when more non-bird species are present in the image. Despite being monosemantic, the magnitude of the activation does not correspond to the "strength" or prevalence of the concept's presence. Conversely, a feature may be monotone and polysemantic—for instance, if higher values indicate a larger presence of both birds and cats. While $k$-SAEs disentangle dense representations into useful concepts, we additionally introduce a novel loss term to explicitly encourage monotonicity.

## 2.3 MONOTONICITY LOSS

To encourage our $k$-SAE to learn monotone features, we introduce an additional contrastive "peripteral" loss term inspired by Bhatt et al. (2024). Concretely, we sample hEterogeneous and hoMogeneous sets $E, M \subset V$, define a margin function $\Delta(E|M)$ which measures how much more diverse $E$ is than $M$, and instantiate an unweighted FB-function $f(A) = \sum_{i \in [d_{\text{sparse}}]} \log(1 + m_i(A))$. The *monotonicity loss* is defined as

$$\mathcal{L}_{\text{mono}}(E, M) = |\Delta(E|M)| \cdot \log\left(1 + \exp\left(1 - \frac{f(E) - f(M)}{\Delta(E|M)}\right)\right). \tag{3}$$

This loss aligns the sign and magnitude of $f(E) - f(M)$ with the margin $\Delta(E|M)$. Intuitively, heterogeneous sets $E$ (sets with more concepts) should achieve higher values of $f$ than homogeneous sets $M$ (sets with few concepts). Because $f(A) = \sum_i \log(1 + m_i(A))$ is concave, repeatedly increasing already-active features contributes little additional gain; across many sampled pairs, the most consistent way to reduce the loss is for new concepts in $E$ to activate additional features. This biases the SAE toward representations that behave monotonically with respect to concept presence.

In practice, the true concept set is unknown, so we approximate $\Delta(E|M)$ by defining $\Delta(E|M) = \sum_{(x,x') \in M} \langle x, x' \rangle - \sum_{(x,x') \in E} \langle x, x' \rangle$, the difference in summed pairwise similarities computed in the original dense representation space. We sample $E$ uniformly at random, and $M$ is obtained by taking a single element of $E$ and finding its $|E| - 1$ nearest neighbors in $V$. The final SAE is trained with a combination of $\mathcal{L}_{\text{recons.}}$ and $\mathcal{L}_{\text{mono}}$. Refer to Algorithm 1 for pseudocode and additional implementation details, and to Appendix D for further analysis of how low monotonicity loss encourages monotone features.

## 2.4 SUBMODULAR DISTRIBUTION MATCHING

Now that we have provided a framework to learn sparse and monotone features, we turn to the problem of subset selection. Specifically, we seek to maximize a feature-based (FB) function to ensure that the final subset maintains a balanced representation of concepts Bilmes & Bai (2017). This section details our subset selection objective.

**Definition 2.1** (Empirical Distribution). *Given a ground set of indexing $V = [n]$, any subset $A \subseteq V$ and design matrix $Z \in \mathbb{R}_+^{n \times d_{\text{sparse}}}$, we define a histogram based empirical distribution $p(A) \in \Delta^{d_{\text{sparse}} - 1}$, such that $p(A)_i = \frac{m_i(A)}{\sum_{j \in [d_{\text{sparse}}]} m_j(A)}$, where $m_i(A) \triangleq \sum_{j \in A} z_{ji}$.*

**Corollary 2.2** (Target Empirical Distribution). *Given a ground set of indexing $V^{\text{tar}}$ and target design matrix $Z^{\text{tar}}$, we define target empirical distribution $\boldsymbol{p}^{\text{tar}} \in \Delta^{d_{\text{sparse}}-1}$, such that $p_i^{\text{tar}} = \frac{m_i(V^{\text{tar}})}{\sum_{j \in [d_{\text{sparse}}]} m_j(V^{\text{tar}})}$, where $m_i(A) \triangleq \sum_{j \in A} z_{ji}^{\text{tar}}$.*

Given the target empirical distribution $\boldsymbol{p}^{\text{tar}} \in \mathbb{R}_+^{d_{\text{sparse}}}$, our goal is to match it as closely as we can, by selecting examples from our source distribution. That is, we aim to find $A \subseteq V^{\text{source}}$ which minimizes $D_{\text{KL}}(\boldsymbol{p}^{\text{tar}} \| \boldsymbol{p}^{\text{source}}(A))$, where $\boldsymbol{p}^{\text{source}}(A)$ denotes the empirical distribution of source for given $A \subseteq V$, as per definition 2.1. In the following theorem, we show that this problem is an instance of maximizing a Difference of Submodular (DS) functions (Iyer & Bilmes, 2012).

**Theorem 2.3** (Distribution Matching as DS Maximization). *Given the target and the source empirical distribution, minimizing $D_{\text{KL}}(\boldsymbol{p}^{\text{tar}} \| \boldsymbol{p}^{\text{source}}(A))$ is an instance of optimizing a Difference of Submodular Functions, where $m_i(A) = \sum_{j \in A} z_{ji}^{\text{source}}$.*

$$A^* = \underset{A \subseteq V}{\operatorname{argmin}} \, D_{\text{KL}}(\boldsymbol{p}^{\text{tar}} \| \boldsymbol{p}^{\text{source}}(A)) \tag{4}$$

$$= \underset{A \subseteq V}{\operatorname{argmax}} \sum_{i \in [d_{\text{sparse}}]} p_i^{\text{tar}} \log(m_i(A)) - \log\left(\sum_{j \in [d_{\text{sparse}}]} m_j(A)\right) \triangleq \underset{A \subseteq V}{\operatorname{argmax}} \, g(A) \tag{5}$$

*Proof.* Please refer to the Appendix E. □

As *eq.* (5) is an instance of DS-maximization, there exists no polynomial time approximate algorithm, unless $P = NP$. Since direct optimization is intractable, we instead maximize a lower bound on $g(A)$ obtained by upper-bounding $\log\left(\sum_{j \in [d_{\text{sparse}}]} m_j(A)\right)$. While tight submodular lower bounds follow from the existence of modular upper bounds (Nemhauser et al., 1978), computing them is prohibitive for large ground sets ($n \sim$ millions). With a budget $b$ in subset selection, we thus consider cardinality-constrained maximization and employ an approximation that, when optimized via accelerated greedy methods (Minoux, 2005; Mirzasoleiman et al., 2014), matches the performance of modular upper-bounds-based algorithms (Iyer & Bilmes, 2012).

**Lemma 2.4.** *Assuming sparse feature map $h : \mathcal{X} \to \mathcal{Z}$ is such that $\|h\|_\infty \le \beta$ and given that $h$ is an instance of $k$-SAE, if $\hat{g}(A) \triangleq \sum_{i \in [d_{\text{sparse}}]} p_i^{\text{tar}} \log(m_i(A)) - \log(k\beta|A|)$, then $\hat{g}(A) \le g(A)$. In many scenarios where subset selection is subject to a cardinality constraint, i.e., $|A| \le b$ for a given budget $b$, the quantity $\hat{g}(A)$ satisfies the submodular lower bound $\hat{g}(A) \ge \sum_{i \in [d_{\text{sparse}}]} p_i^{\text{tar}} \log(m_i(A)) - \log(k\beta b)$.*

*Proof.* Please refer to the Appendix E. In Appendix F, we validate this approach on a Gaussian mixture dataset, showing that the omission of the term $\log(k\beta b)$ does not degrade performance. □

To encourage $\beta$ to be low, we use an activity regularization term as discussed in Section 2.2. To guarantee that the submodular function remains monotone, non-decreasing, and normalized (i.e., it evaluates to 0 on the empty set), we replace $\log(\cdot)$ with $\log(1 + \cdot)$ in the lemma above. This modification has negligible effect in practice because $m_i(A) \gg 1$, as each $m_i(A)$ aggregates millions of samples.

## 2.5 COMBINING DISTRIBUTION MATCHING OBJECTIVE WITH QUALITY MEASURES

While maximizing the above objective helps align the feature distribution of the selected subset with that of the target, it only considers image features. However, effective data filtering must also account for the semantic alignment between each image and its corresponding caption. Prior work has proposed various ways to quantify this alignment—typically by computing the dot product between the image and text embeddings (Gadre et al., 2023; Wang et al., 2024b; Kim et al., 2024; Fang et al., 2024; Maini et al., 2024)—and we aim to incorporate this into our distribution matching framework.

A key advantage of using a submodular objective is that it remains submodular even when combined with a modular function (Bilmes, 2022). Therefore, a simple objective that combines distribution matching with a metric that measures quality would be $f(A) \triangleq \hat{g}(A) + \sum_{a \in A} q(a)$ where $\hat{g}(A)$

is the previously defined distribution matching objective and $q(a)$ measures how well a caption corresponds to image $a$. However, this solution has a few shortcomings. First of all, the gain of the overall objective can be expressed as $f(v|A) = \hat{g}(v|A) + q(v)$ for $v \in V \setminus A$ but only $g(v|A)$ decreases as $|A|$ gets larger. Therefore, $q(v)$ may dominate the value of $f(v|A)$ which diminishes the overall objective's ability to jointly consider distributional similarity and quality. Furthermore, existing measures of quality are noisy and may not be useful in expressing finegrained preferences. In other words, $q(i) > q(j)$ does not necessarily imply that $i$ is preferable to $j$ when $q(i) - q(j)$ is small. Therefore, we devise a novel technique to combine a quality score with our distribution matching objective that (1) improves the overall objective's ability to consider both functions during selection and (2) only uses coarse-grained preferences of the quality score.

Instead of using $q$ directly, we create a feature-based function based on a quantized version of $q$ which can be expressed as follows:

$$q'(A) = \sum_{i \in [\ell]} u_i \log \left( 1 + \sum_{j \in A} \mathbb{I}[q(j) \in [b_{i-1}, b_i)] \right) \tag{6}$$

where $\{b_0, b_1, \ldots, b_\ell\}$ denote bin edges, such that bin $i$ corresponds to the interval $[b_{i-1}, b_i)$ and $u_i$ is a weight that expresses the degree of preference we have for bin $i$. Note that since $q'$ has a form similar to $g$, it is easy to show that maximizing $q'(A)$ is minimizing the KL-divergence to a distribution $\mathbf{u}$ if $\sum_{i \in \ell} u_i = 1$.

**Overall Objective** Since the optimization is cardinality constrained, for distribution matching, we can use $\sum_{i \in [d_{\text{sparse}}]} p_i^{\text{tar}} \log(m_i(A))$ as a proxy, since $\log(k\beta|A|)$ is constant. We can now combine the distribution matching with a quantized version of the quality score, which we call SDM (Submodular Distribution Matching).

---

**Definition 2.5** (SDM). *Given ground sets and design matrices $V^{\text{source}}$, $\mathbf{Z}^{\text{source}}$, modular function $m_i^{\text{source}}(A) \triangleq \sum_{j \in A} z_{ji}^{\text{source}}$ for any subset $A \subseteq V^{\text{source}}$ and $i \in [d_{\text{sparse}}]$ (similarly for the target).*
*Let the target empirical distribution be defined as $\boldsymbol{p}^{\text{tar}} \in \Delta^{d_{\text{sparse}}-1}$, where*

$$p_i^{\text{tar}} \triangleq \frac{m_i^{\text{tar}}(V^{\text{tar}})}{\sum_{j \in [d_{\text{sparse}}]} m_j^{\text{tar}}(V^{\text{tar}})}$$

*Furthermore, given $\ell$ bins $\{b_0, b_1, \ldots, b_\ell\}$ over a quality metric $q$, along with bin weight vector $\boldsymbol{u}$, a trade-off parameter $\lambda \in [0, 1]$, and a selection budget $b$, SDM optimizes:*

$$\underset{\substack{A \subseteq V^{\text{source}} \\ |A|=b}}{\arg\max} \ \lambda \underbrace{\sum_{i \in [d_{\text{sparse}}]} p_i^{\text{tar}} \log\big(1 + m_i^{\text{source}}(A)\big)}_{\text{distribution matching}}$$

$$+ (1-\lambda) \underbrace{\sum_{i \in [\ell]} u_i \log\Big(1 + \sum_{j \in A} \mathbf{1}\{q(j) \in [b_{i-1}, b_i)\}\Big)}_{\text{quality weighting}} \tag{7}$$

---

## 3 Experiments

In our experiments, our primary goal is to determine the most effective subset selection strategy for training a CLIP-style model in the DataComp-medium benchmark (Gadre et al., 2023).

### 3.1 Setup

**Datasets** We use the image-caption dataset associated with the DataComp-medium benchmark (128M image-caption pairs) for all training. All subset selection strategies in this section are designed to identify the best subset from DataComp-medium for training a CLIP-style model. The resulting models are evaluated based on zero-shot performance on a suite of 38 zero-shot tasks proposed by Gadre et al. (2023) which includes ImageNet-1K (Deng et al., 2009), several classification datasets from VTAB (Zhai et al., 2019), and retrieval benchmarks such as MS-COCO (Chen et al., 2015).

**CLIP Training**   We adopt the DataComp-medium training configuration for all experiments. Following the selection of a subset via a given filtering approach, training is conducted using a CLIP-B/32 model under a standardized compute regime where the **total number of training steps is fixed to 128M, regardless of dataset size.** Each training run requires 40 A100 hours.

**SAE Configuration**   We train an SAE on the CLIP ViT-L/14 image embeddings ($d_{\text{in}} = 768$), with $d_{\text{sparse}} = 98,304$. The SAE is trained on the unfiltered DataComp-medium dataset; *no external data sources are used*. Note that the image encoder parameters are frozen, so we are able to train the SAE on precomputed image embeddings.

**Submodular Objective and Optimization**   By default, our submodular objective uses negCLIPLoss (Wang et al., 2024b) as a measure of quality. We quantize this into three equally sized bins, and assign weights of 0, 0.01, 0.99 to the low, medium, and high values respectively. We use the training set of ImageNet-1K as the target distribution unless otherwise stated. To maximize this objective, we use the stochastic greedy algorithm (Mirzasoleiman et al., 2014) (shown in Algorithm 2) with $\epsilon = 0.001$. We run stochastic greedy 5 times to select subsets and compute the intersection between them to get the final subset (see Appendix K for an ablation on this). Selecting a subset of 25M samples from a pool of 128M takes approximately 1 hour on a CPU using stochastic greedy with Bilmes (2026); the total time required to compute the final summary aggregated over 5 stochastic greedy results is 5 hours though this can be fully parallelized across separate CPU's.

**Computational Cost**   Overall, our method requires  60 A100 hours to compute the initial features, 5 A100 hours to compute the quality scores,  15 A100 hours to train the SAE, and  5 CPU hours to run stochastic greedy. Importantly, both feature extraction (a shared bottleneck across all methods) and NegCLIPLoss scoring scale linearly with the size of the candidate pool. In contrast, the SAE training cost can remain fully independent of the pool size. Training CLIP on DataComp-medium requires 40 A100 hours.

## 3.2   MAIN RESULTS

Table 1: **Main Results.** Performance of filtering strategies on medium-scale evaluation tasks. **Bold** values indicate the best in each column, and underlined values indicate the second best. For all methods, we report results at their best-performing data fraction.

| Filtering Strategy | Size | IN1K | IN1K Shifts | VTAB | Retrieval | Avg |
|---|---|---|---|---|---|---|
| No Filter | 128M | 17.6% | 15.2% | 25.9% | 21.9% | 25.8% |
| Basic Filtering (Gadre et al., 2023) | 30M | 22.6% | 19.3% | 28.4% | 25.1% | 28.5% |
| Text-Based (Gadre et al., 2023) | 31M | 25.5% | 21.5% | 32.8% | 24.9% | 30.7% |
| Image-Based (Gadre et al., 2023) | 29M | 26.8% | 21.3% | 31.9% | 25.6% | 31.2% |
| CLIP-Score (Gadre et al., 2023) | 38M | 27.3% | 23.0% | 33.8% | 25.1% | 32.8% |
| Image-Based ∩ CLIP-Score (Gadre et al., 2023) | 14M | 29.7% | 23.9% | 34.6% | 23.1% | 32.8% |
| D2 Pruning  (Maharana et al., 2024) | 26M | 24.1% | 20.6% | 30.6% | 19.6% | 29.8% |
| negCLIPLoss (NCL)  (Wang et al., 2024b) | 33M | 28.8% | 23.8% | 35.4% | 25.3% | 34.4% |
| NCL ∩ NormSim (IN1K)  (Wang et al., 2024b) | 22M | 32.8% | 26.8% | 36.2% | 26.5% | 35.3% |
| NCL ∩ NormSim (Target)  (Wang et al., 2024b) | 22M | 32.7% | 26.5% | 37.5% | 26.5% | 35.7% |
| SDM (ours) | 18M | **35.2%** | **27.1%** | **38.6%** | **26.8%** | **36.4%** |

In this section, we focus on the setting where only a single model (specifically CLIP ViT-L/14) is available for generating embeddings. We argue that this setting is the most general because (1) training a new dedicated model for data selection for a specific domain is often prohibitively expensive—sometimes costing more than training the primary model itself (Fang et al.,

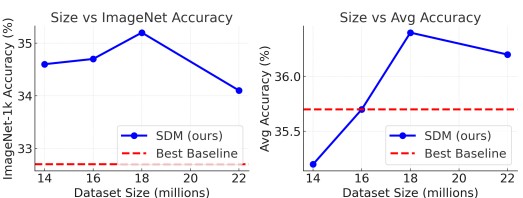

Figure 4: **Different Sizes**. We test SDM at different sizes to the best baseline (NCL ∩ NormSim) which uses 22M samples.

2024; Kim et al., 2024), and (2) approaches
that score data using multiple embedding sources or models (Maini et al., 2024; Yu et al., 2023) are not always feasible, especially in domains where such models are unavailable, such as biomedical image/caption datasets (Ikezogwo et al., 2023; Lozano et al., 2025).

In Table 1, we evaluate several baselines. **CLIP-Score** (Gadre et al., 2023) selects the image-caption pairs whose corresponding image and text embeddings have the highest dot product. **Image-Based** (Gadre et al., 2023) performs k-means clustering on the image embeddings and selects samples which belong to clusters that are close to embeddings of training images from ImageNet. **Image-Based ∩ CLIP-Score** Gadre et al. (2023) simply takes the set intersection between the sets selected by the two methods. **D2 Pruning** (Maharana et al., 2024) constructs an undirected graph initialized with CLIP scores and selects data by jointly optimizing for sample difficulty and diversity. **NCL** (Wang et al., 2024b) is similar to CLIP-Score but assesses sample quality based the CLIP loss (Radford et al., 2021) - a batch wise measure that considers semantic alignment and specificity. **NormSim** measures how similar an image is to a target dataset in terms of p-norm. By default, we always consider $p = \infty$ which was the highest performing variant in Wang et al. (2024b), and consider two target datasets (1) the Imagenet-1k training set or (2) Target which is the training sets of 24 different downstream tasks. **NCL ∩ NormSim** combines the two prior approaches by taking a set intersection of the resulting subsets. This method was previously the state of the art data selection technique among those that only use CLIP ViT-L/14 (Wang et al., 2024b).We find that SDM consistently delivers substantial performance improvements over state-of-the-art data selection methods. On ImageNet-1K, SDM outperforms both variants of NCL ∩ NormSim by 2.5% on ImageNet-1K, and achieves a 0.7% gain in average performance.

Notably, in Figure 4, we find that our method is especially effective at smaller dataset scales: with 33% fewer samples, SDM still outperforms NCL ∩ NormSim by nearly 2% on ImageNet. We attribute this advantage to SDM's ability to reduce redundancy in the selected subsets.

## 3.3 ALTERNATE BACKBONES AND QUALITY SCORES

NormSim (Wang et al., 2024b) and SDM are both target-aware data selection methods that can be applied with different feature extractors and quality scoring functions. Figure 5 compares their performance across several backbone architectures (CLIP ViT-L/14, ViT-B/32, and DFN-P) and quality scores (CLIPScore (Gadre et al., 2023) and NegCLIPLoss).

In each case, we assume access to a single embedding model, as in the previous section. Across all backbones and scoring functions, SDM consistently outperforms NormSim, both on ImageNet and in the

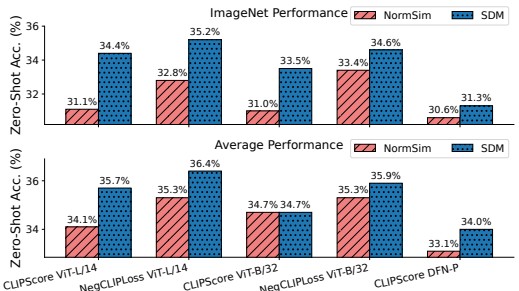

Figure 5: **SDM vs. NormSim**. We test both SDM and NormSim with different quality scores and backbones, and find that SDM consistently achieves better zero-shot performance.

average performance. This demonstrates that distribution-aware selection is more effective than evaluating samples individually.

## 3.4 IMPACT OF MONOTONICITY LOSS

We study the effect of adding the loss term $\mathcal{L}_{\text{mono}}$ to the standard reconstruction loss $\mathcal{L}_{\text{recons.}}$ when training a $k$-SAE. We introduce two metrics to evaluate SAE quality: the Monosemanticity Score (MS Score) and the Monotonicity Score (MT Score). **MS Score** captures semantic consistency of neurons (Zhang et al., 2025). For each neuron $i \in [d_{\text{sparse}}]$, we compute the average pairwise cosine similarity among images activating that neuron, and then average over all neurons. A higher score indicates that images activating the same neuron are semantically aligned. **MT Score** measures how well features behave like "counts." For each neuron $i$, we sort images by activations $Z[:, i]$, yielding permutation $\sigma_i$ with $\sigma_i[1]$ denoting the strongest activation. If $n$ is the dataset size, the neuron-level score is $\frac{1}{n-1} \sum_{j=1}^{n-1} < x_{\sigma_i[j]}, x_{\sigma_i[j+1]} >$. We average the neuron-level score across all neurons, to report the MT Score. Intuitively, adjacent images with similar activations for feature $i$ should also

exhibit similar concept counts, resulting in higher similarity. Finally, we evaluate the impact of $\mathcal{L}_{\mathrm{mono}}$ within SDM, following the same procedure as before but performing distribution matching with a different set of features. We report both zero-shot ImageNet-1K accuracy and the average zero-shot accuracy across all 38 evaluation tasks. Please refer to Appendix I for more details.

Table 2: Impact of $\mathcal{L}_{\mathrm{mono}}$ on SAE training; MS: Monosemanticity Score (↑), MT: Monotonicity Score (↑).

| | DataComp Small | | ImageNet-1K | | Accuracy | |
|---|---|---|---|---|---|---|
| | MS Score | MT Score | MS Score | MT Score | ImageNet-1K | Avg |
| $\mathcal{L}_{\mathrm{recons.}}$ | 0.55 | 0.60 | 0.58 | 0.60 | 34.80% | 35.00% |
| $\mathcal{L}_{\mathrm{recons.}} + \mathcal{L}_{\mathrm{mono}}$ | **0.62** | **0.66** | **0.63** | **0.65** | **35.20%** | **36.40%** |

In Table 2, we show that adding $\mathcal{L}_{\mathrm{mono}}$ yields consistent improvements across all metrics. Although our study focuses on applying SAEs to data selection, the substantial gains in both MS and MT Scores highlight the potential relevance of this approach to the broader interpretability community.

### 3.5 COMPONENT-WISE ABLATION

In this section, we disentangle the contributions of two components in our approach: (1) using SAE-derived sparse features, and (2) using a submodular objective. To isolate the effect of using sparse feataures, we construct the feature-based objective directly on dense CLIP embeddings (after applying a ReLU to ensure nonnegativity). To isolate the effect of submodularity, we remove the $\log()$ function from both components in Theorem 2.5, which eliminates diminishing returns. We report results on ImageNet-1K as well as the average performance across 38 downstream datasets in Table 3. The results demonstrate the using sparse features is critical to downstream performance - using dense CLIP features results in a 12% drop in ImageNet performance and 9% overall. We also find that removing the concave function substantially diminishes overall performance, by 1.3%.

Table 3: **Effect of Sparse Features and Submodularity.** Performance on ImageNet-1K (left) and the average over 38 datasets (right) under two ablations: (i) using dense CLIP features instead of SAE-derived sparse features, and (ii) removing the concave log term that induces submodularity.

| | ImageNet-1K (%) | | | Average Over 38 Datasets (%) | |
|---|---|---|---|---|---|
| | Submodular | No Submodular | | Submodular | No Submodular |
| **Sparse** | **35.2** | 34.6 | **Sparse** | **36.4** | 35.1 |
| **Dense** | 24.1 | 23.7 | **Dense** | 29.1 | 28.9 |

### 3.6 COMBINATION WITH OTHER SELECTION APPROACHES

We also integrate SDM with the highest-performing method on DataComp-medium that publicly releases its indices, namely the NCL + DFN + HYPE ensemble proposed by Wang et al. (2024b). Their approach constructs the dataset by taking the disjoint union of three independently selected subsets; any sample that appears in $k$ subsets is upweighted by being duplicated $k$ times in the final collection. To incorporate SDM, we define a submodular objective that encourages quality in a manner analogous to our earlier

Table 4: **Results w/ External Models.** Performance of filtering strategies on ImageNet and overall average when using external data and models. **Bold** = best, underline = second best.

| Filtering Strategy | IN1K | Avg |
|---|---|---|
| TMARS + SSFT (Maini et al., 2024) | 33.8% | 36.2% |
| HYPE (Kim et al., 2024) | 34.6% | 37.3% |
| DFN (Fang et al., 2024) | 37.1% | 37.3% |
| NCL + DFN + HYPE (Wang et al., 2024b) | 38.2% | 38.8% |
| M-FLYT + SCS (Shechter & Carmon, 2025b) | **40.1%** | 37.7% |
| Metagradient Descent (Engstrom et al., 2025) | 27.0% | **40.2%** |
| SDM + (Wang et al., 2024b) (ours) | 39.2% | 39.2% |

formulation and apply the same upweighting scheme. To further boost performance, we instantiate SDM objectives using SAE features derived from two embedding models—CLIP ViT-L/14 and DINOv2. Full details of this procedure are provided in Appendix J.

As shown in Table 4, this hybrid approach achieves substantial gains over the base method and ranking **2nd** based on average performance out of all methods on the DataComp-medium leaderboard. The only method outperforming SDM is Metagradient Descent (Engstrom et al., 2025), which is prohibitively expensive and attains $12.2\%$ lower zero-shot accuracy on ImageNet-1K. Please refer to Appendix B, which discusses how *SDM requires over 5x less GPU hours than the Metagradients approach.*

## 4 RELATED WORK

**Targeted Data Selection.** Targeted data selection is a broader field that has been explored in other settings as well. Wallingford et al. (2023); Udandarao et al. (2023) propose similarity based data selection methods for few-shot adaptation of VLMs, which also evaluate the utility of each sample independently. Some recent work also explores using submodularity to select samples that are diverse and relevant to a target (Das et al., 2025; Kothawade et al., 2021; Kumari et al., 2024; Agarwal et al., 2024) in various contexts. However, these methods are difficult to scale due to their reliance on pairwise similarity matrices. Finally, there has also work that uses the regularized optimal transport objective (Liu et al., 2024) in addition to kernel density-based estimation to promote diversity. However, these methods cannot include a quality score trade-off and can still be $\mathcal{O}(n \log n)$, even ignoring the subset selection cost, where $n$ is the size of the dataset to select from. *Our work approximates the distribution matching objective with a tractable submodular function, and can easily handle additional quality score(s).*

**SAE's in Data Selection.** Two recent works have explored the use of SAE's for data selection for supervised finetuning of LLMs. Yang et al. (2025) introduced the first approach in this direction, proposing *SAE-GreedSelect*, which greedily builds a subset by repeatedly selecting the sample that covers the largest number of previously uncovered concepts. However, this strategy is ill-suited to our setting, where $d_{\text{sae}}$ is much smaller than the target summary size, causing the pool of concepts to be exhausted prematurely. Moreover, Yang et al. (2025) do not provide mechanisms to (1) combine diversity and quality—essential when working with highly uncurated multimodal datasets—or (2) assign relative importance to different concepts, both of which are addressed by SDM. More recently, Ma et al. (2025) proposed a method similar to NormSim (Wang et al., 2024b), for a task-aware selection method that uses SAE-derived representations to construct a more robust similarity metric.

## 5 CONCLUSION AND FUTURE WORK

Overall, we propose SDM, a novel submodular framework for filtering multimodal datasets using a distribution-matching objective based on SAE features, combined with a quality-based objective. When filtering the DataComp-medium pool, and only using CLIP embeddings, SDM achieves SOTA accuracy on both ImageNet-1K and average performance across 38 downstream tasks. On the full benchmark, SDM delivers performance within 1% of the SOTA while using over **5×** fewer GPU hours than the leading approach. There are several promising directions for future work. First, our framework is modality/domain-agnostic and can be readily applied to other tasks which we plan to explore in the future. Second, while our main experiments fixed the target dataset to ImageNet, we include in Appendix K an ablation in which we vary the target dataset. This analysis shows that performance can be sensitive to the choice of target, suggesting that careful selection of the target distribution is important. Exploring principled strategies for target selection, such as using strategies inspired by data mixture optimization (Xie et al., 2023), remains an exciting direction for future work.

## 6 ACKNOWLEDGEMENTS

This work is supported by NSF Grant Nos. IIS-2106937 and IIS-2148367.

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

## A    TABLE OF NOTATION

Here we provide a list of notations we used throughout this work.

| Notation | Description |
|---|---|
| $V = V^{\text{source}} = [n]$ | Source data indices (ground set) |
| $V^{\text{tar}}$ | Target data indices |
| $\mathcal{X} \subseteq \mathbb{R}_+^{d_{\text{in}}}$ | Input image feature space |
| $\mathcal{Z} \subseteq \mathbb{R}_+^{d_{\text{sparse}}}$ | Sparse representation space |
| $\boldsymbol{Z}^{\text{source}}$ | Source design matrix |
| $\boldsymbol{Z}^{\text{tar}}$ | Target design matrix |
| $\boldsymbol{Z}_j^{\text{source}} \triangleq h^{\text{enc}}(\boldsymbol{x}_j^{\text{source}})^T$ | Encoded rep. of $j$-th source input |
| $h^{\text{enc}} : \mathcal{X} \to \mathcal{Z}$ | Sparse autoencoder encoder |
| $\Delta^{d-1}$ | $d$-dimensional simplex |
| $\mathbb{I}[\cdot]$ | Indicator function |
| $\phi$ | Concave function |
| $m(A)$ | Modular function on $A \subseteq V$ |
| $\mathcal{I}$ | Image space |
| $\mathcal{T}$ | Caption space |
| $\mathcal{D}^{\text{source}}$ | Source image-caption pairs |
| $q(k)$ | Quality score for index $k$ |
| $\mathcal{D}^{\text{tar}}$ | Target image set |
| $\mathcal{D}^{\text{train}}$ | Selected training subset |

Table 5: Summary of Notations

## B    ADDITIONAL RELATED WORK

**DataComp**    The DataComp paper introduces a benchmark that fixes the training and evaluation procedures for VLMs to measure the effect of data filtering techniques on model performance Gadre et al. (2023). Since then, many new data filtering techniques have emerged and can broadly be categorized into two types: **(1)** approaches that rely exclusively on off-the-shelf CLIP embeddings while refining how those embeddings are employed Gadre et al. (2023); Maharana et al. (2024); Wang et al. (2024b); **(2)** methods that use curated external data sources to either enhance the CLIP model Fang et al. (2024) itself or incorporate wholly different external models to guide the data-selection process Kim et al. (2024); Yu et al. (2023); Maini et al. (2024). Generally, the latter outperform the former but demand substantially more development resources—often exceeding those required to train the VLM itself. *Our work falls in the former category, but can be easily combined with methods in the latter.*

**Comparison with Metagradients**    Engstrom et al. (2025) frames data-subset selection as gradient-based hyperparameter tuning, where each datapoint's weight in the loss is treated as a hyperparameter. However, this approach requires *"backpropagation through the entire training sequence"*, which naively incurs a prohibitive memory cost of $\mathcal{O}(T)$, where $T$ is the number of training steps. To mitigate this, the paper introduces a $\mathcal{O}(k \log T)$ space algorithm using a lazy $k$-ary tree, akin to the island algorithm (Binder et al., 1997). Beyond memory, the method still necessitates substantial computation: training on the DataComp dataset for 40 steps (Fig. 6 from their paper, assuming they require a similar number of steps for DataComp Medium, since they only report experiments on the DataComp Small dataset) translates to $40 \times 5 = 200$ A100-hours on NVIDIA A100 GPUs, assuming 5 A100-hours per metagradient step. In contrast, our approach trains at most 2 SAEs, requiring only 40 hours. ***Thus, SDM is approximately 5× faster than the Metagradients-based approach.***

## C  PRIMER ON SUBMODULAR FUNCTIONS

In this section, we formally describe submodular functions. For a given ground set $V = [n] \triangleq \{1, \ldots, n\}$, a set function $f : 2^V \to \mathbb{R}$ is submodular if and only if it satisfies $f(A \cup \{v\}) - f(A) \geq f(B \cup \{v\}) - f(B)$ for subsets $A \subseteq B \subseteq V$ and $v \in V \setminus B$, where $f(v|A) \triangleq f(A \cup \{v\}) - f(A)$ is often referred to as gain of adding a new element $v$. In case when $f(v|A) = f(\{v\})$, then the function is referred to as a modular function, and in such case for any $A \subseteq V$, it can be decomposed as $f(A) = \sum_{j \in A} f(\{j\}) + f(\emptyset)$. Diminishing returns allows submodular functions to effectively model notions of diversity and coverage Bilmes (2022). When set functions are monotone non-decreasing ($f(A) \leq f(B), \forall A \subseteq B \subseteq V$) and normalized ($f(\emptyset) = 0$), they are referred to as polymatroid functions. In general, maximizing submodular functions is NP-Hard, however, polymatroid functions can be *approximately*-maximized subject to a variety of constraints from set cardinality, to knapsack, to a bigger and theoretically interesting class of matroid rank constraints. For cardinality constraints, a greedy algorithm achieves an approximation guarantee of $1 - e^{-1}$ (Feige, 1998; Minoux, 2005; Nemhauser et al., 1978); for other constraints, please refer to Badanidiyuru & Vondrák (2014).

## D  ANALYSIS OF MONOTONICITY LOSS

The goal of this section is to show that minimizing the monotonicity loss $\mathcal{L}_{\mathrm{mono}}$ on carefully constructed $(E, M)$ pairs, together with the margin function $\Delta(E \mid M)$, results in learned features that are monotone with respect to a set of concepts.

**Definition D.1** (Monotonicity). *Let $\mathscr{C}$ be a finite set of concepts. For each item $v \in V$ associate a concept vector $\psi(v) \in [0, 1]^{|\mathscr{C}|}$. Given nonnegative features $z \in \mathbb{R}_+^{|V| \times d}$, we say that $z$ is* monotone *with respect to $\mathscr{C}$ if for every $c \in \mathscr{C}$ there exists a nonempty index set $G_c \subseteq [d]$ and a constant $\eta \geq 0$ such that, for any two items $v, v' \in V$ with $\psi_c(v) > \psi_c(v')$,*

$$\sum_{i \in G_c} (z_{vi} - z_{v'i}) \geq \eta(\psi_c(v) - \psi_c(v')).$$

**Remark D.2** (Monotonicity vs. Monosemanticity). *It is important to distinguish monotonicity from monosemanticity. Monotonicity requires that some coordinates consistently increase as the presence of a concept grows, but it does not require a one-to-one mapping between concepts and features. A single feature may still respond to multiple concepts, as long as its activation is nondecreasing in each relevant direction. By contrast, monosemanticity typically implies that each feature should be specialized to a single concept.*

**Definition D.3** (Feature Sums and Aggregator). *For $z \in \mathbb{R}_+^{|V| \times d}$ and subset $A \subseteq V$, define*

$$m_i(A) = \sum_{j \in A} z_{ji}, \quad \forall i \in [d], \qquad f(A) = \sum_{i=1}^{d} \log(1 + m_i(A)).$$

**Definition D.4** (Concept Margin). *For subsets $E, M \subseteq V$ of equal size, define*

$$\Delta(E \mid M) = \|\psi(E) - \psi(M)\|_1,$$

*which measures the total excess of concepts in $E$ compared to $M$.*

**Definition D.5** (Monotonicity Loss). *For a pair $(E, M)$, the contrastive loss is*

$$\mathcal{L}_{\mathrm{mono}}(E, M) = \Delta(E \mid M) \cdot \log\left(1 + \exp\left(1 - \frac{f(E) - f(M)}{\Delta(E|M)}\right)\right).$$

*Given a dataset $\mathcal{T} = \{(E_t, M_t)\}_{t=1}^{n}$, the total loss is*

$$\mathcal{J}(z) = \sum_{t \in \mathcal{T}} \mathcal{L}_{\mathrm{mono}}(E_t, M_t).$$

**Definition D.6** (Per-Concept Partition). *For each concept $c \in \mathscr{C}$, define*

$$\mathcal{T}_c = \{(E_t, M_t) \in \mathcal{T} : E_t, M_t \text{ differ only in concept } c\}.$$

*Let*

$$\Gamma_c = \sum_{t \in \mathcal{T}_c} \Delta(E_t \mid M_t), \qquad \mathcal{J}_c(z) = \sum_{t \in \mathcal{T}_c} \mathcal{L}_{\mathrm{mono}}(E_t, M_t).$$

**Theorem D.7** (Low Loss Implies Average-Case Concept Monotonicity). *Suppose for every $c \in \mathscr{C}$ the per-concept average loss is small:*

$$\frac{1}{\Gamma_c} \mathcal{J}_c(z) \leq \varepsilon.$$

*Then for each c there exists a nonempty $G_c \subseteq [d]$ such that*

$$\frac{1}{\Gamma_c} \sum_{t \in \mathcal{T}_c} \sum_{i \in G_c} \left( m_i(E_t) - m_i(M_t) \right) \geq \tfrac{1}{d}(1 - \varepsilon).$$

*In particular, on average across dataset pairs differing only in c, features in $G_c$ grow proportionally with $\psi_c$, with margin $\eta = \frac{1}{d}(1 - \varepsilon)$.*

*Proof.* Fix $c \in \mathscr{C}$ and write $\Delta_t = \Delta(E_t \mid M_t)$. Define

$$\rho_t(z) = \frac{f(E_t) - f(M_t)}{\Delta_t}, \qquad \Delta_t > 0.$$

Then

$$\mathcal{J}_c(z) = \sum_{t \in \mathcal{T}_c} \Delta_t \cdot \log\left(1 + \exp(1 - \rho_t)\right).$$

Since $\log(1 + e^u) \geq u$, taking $u = 1 - \rho_t$ gives

$$\mathcal{J}_c(z) \geq \sum_{t \in \mathcal{T}_c} \Delta_t(1 - \rho_t) = \Gamma_c - \sum_{t \in \mathcal{T}_c} \Delta_t \rho_t.$$

Dividing by $\Gamma_c$ and applying the loss bound,

$$\frac{1}{\Gamma_c} \sum_{t \in \mathcal{T}_c} \Delta_t \rho_t \geq 1 - \varepsilon. \tag{1}$$

By definition,

$$\rho_t = \frac{f(E_t) - f(M_t)}{\Delta_t}.$$

Thus (1) implies

$$\frac{1}{\Gamma_c} \sum_{t \in \mathcal{T}_c} \left( f(E_t) - f(M_t) \right) \geq 1 - \varepsilon. \tag{2}$$

Now, note that for any $x, y \geq 0$,

$$\log(1 + x) - \log(1 + y) \leq x - y.$$

Applying this coordinatewise,

$$f(E_t) - f(M_t) \leq \sum_{i=1}^{d} \left( m_i(E_t) - m_i(M_t) \right). \tag{3}$$

Combining (2) and (3),

$$\frac{1}{\Gamma_c} \sum_{t \in \mathcal{T}_c} \sum_{i=1}^{d} \left( m_i(E_t) - m_i(M_t) \right) \geq 1 - \varepsilon. \tag{4}$$

Equation (4) states that, averaged over pairs differing only in concept $c$, the total feature mass increases in line with $\psi_c$. Averaging across coordinates,

$$\frac{1}{d} \sum_{i=1}^{d} \left( \frac{1}{\Gamma_c} \sum_{t \in \mathcal{T}_c} \left( m_i(E_t) - m_i(M_t) \right) \right) \geq \tfrac{1}{d}(1 - \varepsilon).$$

Hence at least one coordinate (and thus some nonempty $G_c$) satisfies the same bound, proving the claim. $\qquad \square$

# E PROOFS FROM THE MAIN PAPER

For completeness, we re-state the statement of the theorem

**Theorem E.1** (Distribution Matching as DS Maximization). *Given the target and the source empirical distribution, minimizing $D_{\mathrm{KL}}(\boldsymbol{p}^{\mathrm{tar}} \parallel \boldsymbol{p}^{\mathrm{source}}(A))$ is an instance of optimizing a Difference of Submodular Functions.*

$$A^* = \underset{A \subseteq V}{\operatorname{argmin}} \, D_{\mathrm{KL}}(\boldsymbol{p}^{\mathrm{tar}} \parallel \boldsymbol{p}^{\mathrm{source}}(A)) \tag{8}$$

$$= \underset{A \subseteq V}{\operatorname{argmax}} \sum_{i \in [d_{\mathrm{sparse}}]} p_i^{\mathrm{tar}} \log(m_i(A)) - \log\left(\sum_{j \in [d_{\mathrm{sparse}}]} m_j(A)\right)$$

$$\triangleq \underset{A \subseteq V}{\operatorname{argmax}} \, g(A) \tag{9}$$

*where $m_i(A) = \sum_{j \in A} z_{ji}^{\mathrm{source}}$.*

*Proof.* We begin by expanding the definition of the KL divergence.

$$D_{\mathrm{KL}}(\boldsymbol{p}^{\mathrm{tar}} \parallel \boldsymbol{p}^{\mathrm{source}}(A)) = \sum_{i \in [d_{\mathrm{sparse}}]} p_i^{\mathrm{tar}} \log\left(\frac{p_i^{\mathrm{tar}}}{p^{\mathrm{source}}(A)_i}\right) \tag{10}$$

$$= -H(\boldsymbol{p}^{\mathrm{tar}}) - \sum_{i \in [d_{\mathrm{sparse}}]} p_i^{\mathrm{tar}} \log\left(p^{\mathrm{source}}(A)_i\right) \tag{11}$$

Where $H(\cdot)$ refers to the shannon entropy function. Since entropy of target distribution is not a function of A, we can focus on the second term, and aim to maximize it.

$$\underset{A \subseteq V}{\operatorname{argmin}} \, D_{\mathrm{KL}}(\boldsymbol{p}^{\mathrm{tar}} \parallel \boldsymbol{p}^{\mathrm{source}}(A)) = \underset{A \subseteq V}{\operatorname{argmax}} \sum_{i \in [d_{\mathrm{sparse}}]} p_i^{\mathrm{tar}} \log\left(p^{\mathrm{source}}(A)_i\right) \tag{12}$$

Now we use the fact that

$$p_i^{\mathrm{source}} \triangleq \frac{m_i(A)}{\sum_{j \in [d_{\mathrm{sparse}}]} m_j(A)}$$

Plugging above yields

$$\underset{A \subseteq V}{\operatorname{argmax}} \sum_{i \in [d_{\mathrm{sparse}}]} p_i^{\mathrm{tar}} \log\left(p^{\mathrm{source}}(A)_i\right) = \underset{A \subseteq V}{\operatorname{argmax}} \sum_{i \in [d_{\mathrm{sparse}}]} p_i^{\mathrm{tar}} \log(m_i(A)) - \log\left(\sum_{j \in [d_{\mathrm{sparse}}]} m_j(A)\right)$$

$\square$

**Lemma E.2.** *Assuming a sparse feature map $h : \mathcal{X} \to \mathcal{Z}$ is such that $\|h\|_\infty \le \beta$ and given that $h$ is an instance of $k$-SAE, if $\hat{g}(A) \triangleq \sum_{i \in [d_{\mathrm{sparse}}]} p_i^{\mathrm{tar}} \log(m_i(A)) - \log(k\beta|A|)$, then $\hat{g}(A) \le g(A)$. Moreover, $\hat{g}(A)$ is submodular for any $A \subseteq V$.*

*Proof.* To arrive at this lower bound, first observe that –

$$\sum_{j \in [d_{\mathrm{sparse}}]} m_j(A) = \sum_{j \in [d_{\mathrm{sparse}}]} \sum_{i \in A} z_{ij} \tag{13}$$

$$= \sum_{i \in A} \sum_{j \in [d_{\mathrm{sparse}}]} z_{ij} \tag{14}$$

Since the design matrix is generated using a TopK sparse autoencoder features, for every example $i \in A$, define $\xi_i \triangleq \{k : k \in [d_{\mathrm{sparse}}], z_{ik} > 0\}$; note that $|\xi_i| \le k$. Since $\|h\|_\infty \le \beta$, therefore, $z_{ik} \le \beta$ for all $i \in A$ and $k \in \xi_i$. Therefore, we have the following –

$$\sum_{j\in[d_{\text{sparse}}]} m_j(A) = \sum_{i\in A}\sum_{j\in\xi_i} z_{ij} \tag{15}$$

$$\leq \sum_{i\in A} k\,\beta \tag{16}$$

$$= |A|\,k\,\beta \tag{17}$$

Plugging the above relation in the definition of $\hat{g}(A)$ yields us the desired inequality. $\qquad\square$

## F  Illustration of Distribution Matching

We generate a synthetic dataset by sampling from a mixture of 30 two-dimensional Gaussian components with diverse means and covariances, which defines the source distribution. The target distribution is constructed by selecting 4 of these Gaussians and oversampling from them, concentrating probability mass on a restricted subset of the mixture. To represent the data, the 2D domain is partitioned into a fixed $50 \times 50$ grid, yielding 2,500 bins. Each sample activates only the bin it falls into, and by aggregating counts, we form an empirical distribution over the bins. Figure 6 illustrates both the source and target datasets.

In this setup, the design matrix $\boldsymbol{Z}$ has one-hot entries along the feature axis. It follows that

$$\sum_{j\in[d_{\text{sparse}}]} m_j(A) = \sum_{j\in[d_{\text{sparse}}]}\sum_{i\in A} z_{ij} \tag{18}$$

$$= \sum_{i\in A}\sum_{j\in[d_{\text{sparse}}]} z_{ij} \tag{19}$$

$$= \sum_{i\in A} 1 = |A|. \tag{20}$$

Consequently, the objective function corresponds to Lemma 2.4, with the overall objective:

$$\operatorname*{argmax}_{A\subseteq V} \sum_{i\in[d_{\text{sparse}}]} p_i^{\text{tar}}\log(m_i(A)) - \log\left(|A|\right),$$

where $d_{\text{sparse}} = 2500$. In this setup, the target distribution has an entropy of **5.467**. Optimizing only the first term (excluding $\log(|A|)$) to match the target size results in a sharp decline in KL divergence, achieving **3.07**. In contrast, random subsets of the same size yield a much higher KL divergence, averaging $23.46 \pm 1.90$ across 1000 trials.

Finally, we observe that the supergradient-based approach that optimizes the full difference, including the $\log(|A|)$ term as in Iyer & Bilmes (2012), performs similarly to optimizing only the first term. This demonstrates the robustness of our method, which is also orders of magnitude faster than optimizing the full objective.

## G  Algorithm Blocks

Here, we include pseudocode for the SAE training procedure used in all experiments (Algorithm 1) and the stochastic greedy procedure employed to maximize the final submodular objective (Algorithm 2). To train the SAE (Algorithm 1), we use $|\mathcal{M}| = 50,000$, $k = 128$, $\lambda_1 = 1$, $\lambda_2 = 0.002$ and $\lambda_3 = 10^{-10}$. In Algorithm 2, we use $\epsilon = 10^{-3}$.

## H  Complexity Analysis

Let $N = |V^{\text{source}}|$, $M = |V^{\text{tar}}|$, $b$ is our budget, and $d^{\text{sparse}}$ is the dimensionality of our sparse encodings with at a time only $k$ entries being non-zero. Our algorithm has two main components:

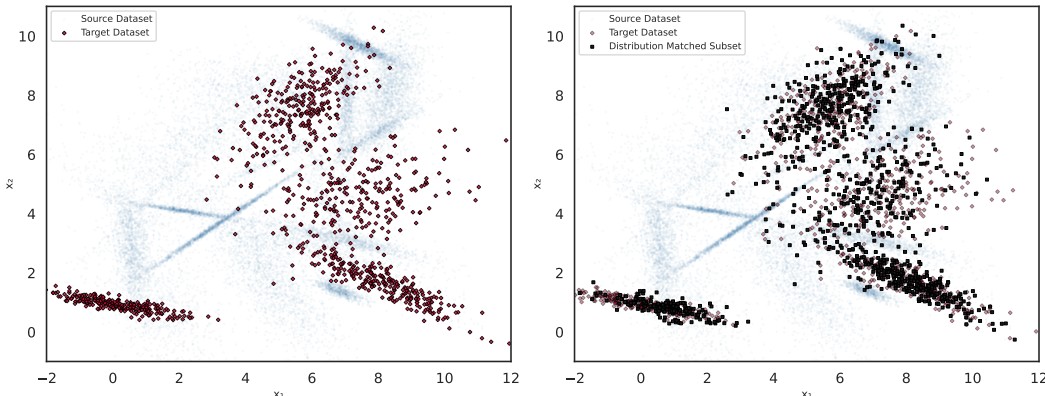

Figure 6: Source dataset (left) and target subset (right) derived from a mixture of Gaussians. We optimize only the first term, as suggested in Lemma 2.4, to obtain a subset that effectively covers the target components.

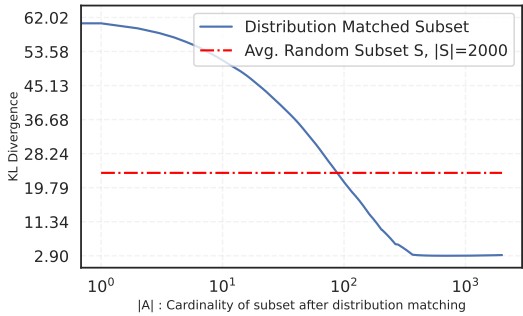

Figure 7: KL divergence reduction over optimization steps, compared to random subsets of size 2000.

(1) generating and storing sparse features, and (2) performing submodular maximization. Sparse feature generation involves simple matrix multiplication. Since each example has only $k$ non-zero entries (128 in our case), the space required to store sparse features is $\mathcal{O}(N)$ and $\mathcal{O}(M)$ for the source and the target, respectively. Our submodular objective combines a binned quality function and feature-based diversity terms. Because the quality function is discretized, it can be stored efficiently using fewer bits than standard floating-point representations. The overall evaluation cost of the objective is $\mathcal{O}(N + d^{\mathrm{sparse}})$. For the maximization step, we employ the stochastic greedy algorithm Mirzasoleiman et al. (2014) (shown in Algorithm 2), which offers a near-optimal approximation guarantee $1 - 1/e - \varepsilon$ in $\mathcal{O}(N \log \frac{1}{\varepsilon})$, instead of $\mathcal{O}(bN)$.

# I  MORE ON MONOTONICITY AND MONOSEMANTICITY SCORES

To evaluate the effect of $\mathcal{L}_{\mathrm{mono}}$, we compare SAEs trained with and without this loss on two datasets: ImageNet-1K and DataComp-Small (a 10% subset of DataComp-Medium). These datasets provide diverse and noisy web-scale images, allowing us to assess robustness.

**Monosemanticity Score**   measures the semantic consistency of neurons (Zhang et al., 2025; Wang et al., 2024a). For each neuron $i \in [d_{\mathrm{sparse}}]$, we compute the average pairwise cosine similarity among images that activate it. A higher score indicates that the neuron consistently responds to semantically aligned images. We plot histograms of these scores for all neurons, comparing the two SAEs.

**Monotonicity Score**   quantifies how well features behave like "counts." For neuron $i$, we sort images by activation values $Z[:, i]$, yielding a permutation $\sigma_i$, with $\sigma_i[1]$ denoting the strongest

---

**Algorithm 1** Train Sparse Autoencoder (SAE)

---

**Require:** Batch list $\mathcal{B}$, set size $k$, buffer size $B$, weights $\lambda = [\lambda_1, \lambda_2, \lambda_3]$, learning rate $\eta$
**Ensure:** Trained SAE $h_\phi$
 1: // Instantiate SAE and buffer for reservoir sampling.
 2: $\mathcal{M} \leftarrow [\,]$
 3: $h_\phi \leftarrow$ INSTANTIATESAE
 4: **for** each batch $b \in \mathcal{B}$ **do**
 5: $\quad \mathcal{M} \leftarrow$ RESERVOIRSAMPLING$(\mathcal{M}, b, B)$
 6: $\quad$ // Compute TopK SAE loss according to eq. 2.
 7: $\quad \mathcal{L}_{\text{SAE}} \leftarrow$ TOPKSAELOSS$(b, h_\phi)$
 8: $\quad \mathcal{L}_{\text{act-reg}} \leftarrow \sum_{x \in b} \|h_\phi(x)\|^2 / |b|$
 9:
10: $\quad$ // Use a random subset as hEterogeneous set.
11: $\quad$ Sample $E \subset b$ such that $|E| = k$
12: $\quad$ Choose $e \in E$ at random
13: $\quad$ // Get $k$-1 nearest neighbor of $e$ from $\mathcal{M}$
14: $\quad$ // Use nearest neighbors as hoMogeneous set.
15: $\quad M \leftarrow \{e\} \cup$ NEARESTNEIGHBOR$(e, \mathcal{M}, k-1)$
16: $\quad$ // Compute set contrastive loss according to eq. 3.
17: $\quad \mathcal{L}_{\text{mono}} \leftarrow$ SETCONTRASTIVELOSS$(E, M, h_\phi)$
18:
19: $\quad \mathcal{L}_{\text{total}} \leftarrow \lambda_1 \cdot \mathcal{L}_{\text{SAE}} + \lambda_2 \cdot \mathcal{L}_{\text{mono}} + \lambda_3 \cdot \mathcal{L}_{\text{act-reg}}$
20: $\quad$ // Update model using gradient descent.
21: $\quad h_\phi \leftarrow$ GRADIENTDESCENT$(h_\phi, \mathcal{L}_{\text{total}}, \eta)$
22: **end for**
23: **return** $h_\phi$

---

**Algorithm 2** Stochastic-Greedy

---

**Require:** $f : 2^V \rightarrow \mathbb{R}_+, k \in \{1, \ldots, n\}, \varepsilon$
**Ensure:** A set $A \subseteq V$ satisfying $|A| \leq k$
 1: $A \leftarrow \emptyset$
 2: **for** $i \leftarrow 1$ to $k$ **do**
 3: $\quad R \leftarrow$ a random subset of size $\left\lceil \frac{n}{k} \log \frac{1}{\varepsilon} \right\rceil$ obtained by sampling $s$ random elements from $V \setminus A$
 4: $\quad a_i \leftarrow \arg\max_{a \in R} \Delta(a \mid A)$
 5: $\quad A \leftarrow A \cup \{a_i\}$
 6: **end for**
 7: **return** $A$

---

activation. If $n$ is the dataset size, the score is

$$\frac{1}{n-1} \sum_{j=1}^{n-1} \langle x_{\sigma_i[j]}, x_{\sigma_i[j+1]} \rangle.$$

Intuitively, adjacent images in this ordering should exhibit similar concept counts, leading to higher similarity. We again plot histograms of these scores across neurons.

In the main paper, we aggregate these histograms by taking the mean, as reported in Table 2.

## J  SDM WITH DATACOMP METHODS THAT USE EXTERNAL MODELS

In this section, we describe how we combine SDM with the ensemble method proposed by Wang et al. (2024b). Overall, we adapt the quality function to incorporate ..., we include per-sample weights, and finally add another SAE trained on DINOv2 (Oquab et al., 2024) features.

Their approach constructs three distinct subsets of DataComp, each selected by a different strategy. Let $\mathcal{D}_k$ denote the set of samples that appear in exactly $k$ of the three subsets, where $k \in 0, 1, 2, 3$.

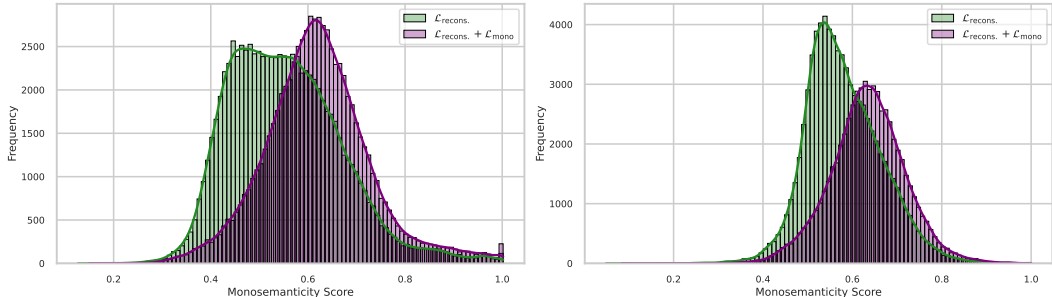

Figure 8: MS Score for DataComp-Small            Figure 9: MS Score for ImageNet-1K

Figure 10: Impact of $\mathcal{L}_{\mathrm{mono}}$ on Monosemanticity Score (higher is better). On both datasets, $\mathcal{L}_{\mathrm{mono}}$ improves semantic consistency, enhancing interpretability.

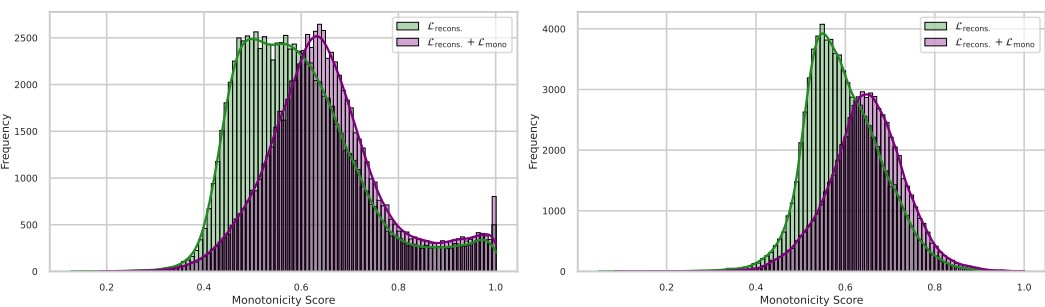

Figure 11: MT Score for DataComp-Small            Figure 12: MT Score for ImageNet-1K

Figure 13: Impact of $\mathcal{L}_{\mathrm{mono}}$ on Monotonicity Score (higher is better). Across both datasets, $\mathcal{L}_{\mathrm{mono}}$ improves the count-like interpretability of features, making them better suited for feature-based submodular functions.

That is, $\mathcal{D}_k$ contains all samples jointly selected by $k$ of the methods. We then define a new quality function, analogous to the one introduced in Section 2.5, as

$$q(A) \triangleq \sum_{i=0}^{3} u_i, \log\left(1 + \sum_{j \in A} \mathbb{I}[j \in \mathcal{D}_i]\right). \tag{21}$$

Here, the weights $u_i$ determine the relative preference for samples selected by multiple methods. In particular, we set $u_i = i$ which encourages the selection procedure to favor samples that are chosen by a larger number of the ensemble methods.

To further strengthen SDM, we instantiate this quality function using features from two independently trained SAEs: one based on CLIP ViT-L/14 embeddings and another based on DINOv2 embeddings. The SAE trained on DINOv2 uses the same hyperparameters as the original SAE.

Finally, we note that all competitive methods in DataComp use some form of sample reweighting, where samples may appear multiple times in the final dataset. Since SDM by design selects subsets rather than multisets, we adopt the reweighting scheme of Wang et al. (2024b), repeating each sample in $\mathcal{D}_k$ $k$ times.

# K    ADDITIONAL ABLATIONS

**Removing Intersection over 5 runs**    For each step of the stochastic greedy algorithm, we randomly sample a subset of $V \setminus A$ (where the size of the subset is controlled by the parameter $\epsilon$) and evaluate the gain only on these. Occasionally, we may get unlucky and random sampling will select a very poor candidate set, leading to the addition of a bad sample. To mitigate this, we take the intersection of multiple stochastic greedy runs, which effectively filters out such suboptimal selections. Moreover, this procedure is parallelizable across cores, allowing us to perform several runs in parallel with minimal computational overhead. In Table 6, we show the comparison between a single run and the intersection over 5 runs in downstream performance.

Table 6: **Effect of Intersecting Multiple Runs** Comparison of a single run versus intersection over 5 independent stochastic greedy runs of maximizing the SDM objective.

| Method | IN1K (%) | Avg. (%) |
|---|---|---|
| Single run | 33.5 | 36.0 |
| Intersection over 5 runs | **35.2** | **36.4** |

**Modifying the Target**    The choice of target distribution plays a central role in the effectiveness of SDM. Following prior work (Wang et al., 2024b; Shechter & Carmon, 2025a; Gadre et al., 2023), our main experiments use ImageNet-1K as the target distribution for subset selection. However, we also tested a broader target distribution constructed by concatenating the training splits of 24 downstream datasets (including ImageNet), as done in Wang et al. (2024b). Interestingly, this reduced overall performance as shown in Table 7. We hypothesize that this decrease stems from (1) severe concept imbalance when merging datasets of different sizes and concept distributions, and (2) inclusion of low-quality datasets (e.g., CIFAR-10), whose samples degrade performance on others. Developing more principled strategies for constructing larger target distributions (e.g., via data mixture optimization (Xie et al., 2023)) remains an interesting direction for future work.

Table 7: **Modifying the Target** Comparison of performance when changing the target from ImageNet-1K to a pool of 24 downstream datasets as done in Wang et al. (2024b).

| Target | IN1K | IN1K Shifts | VTAB | Retrieval | Avg |
|---|---|---|---|---|---|
| ImageNet-1K | **35.2** | **27.1** | **38.6** | **26.8** | **36.4** |
| 24 datasets | 33.3 | 26.7 | 37.1 | 26.7 | 35.8 |

## L    ADDITIONAL SAE VISUALIZATIONS

In this section, we present visualizations of several additional neurons randomly selected from the SAE trained on CLIP embeddings. For each neuron, we display the top 5, middle 5, and bottom 5 images with nonzero activations, ordered in descending activation value.

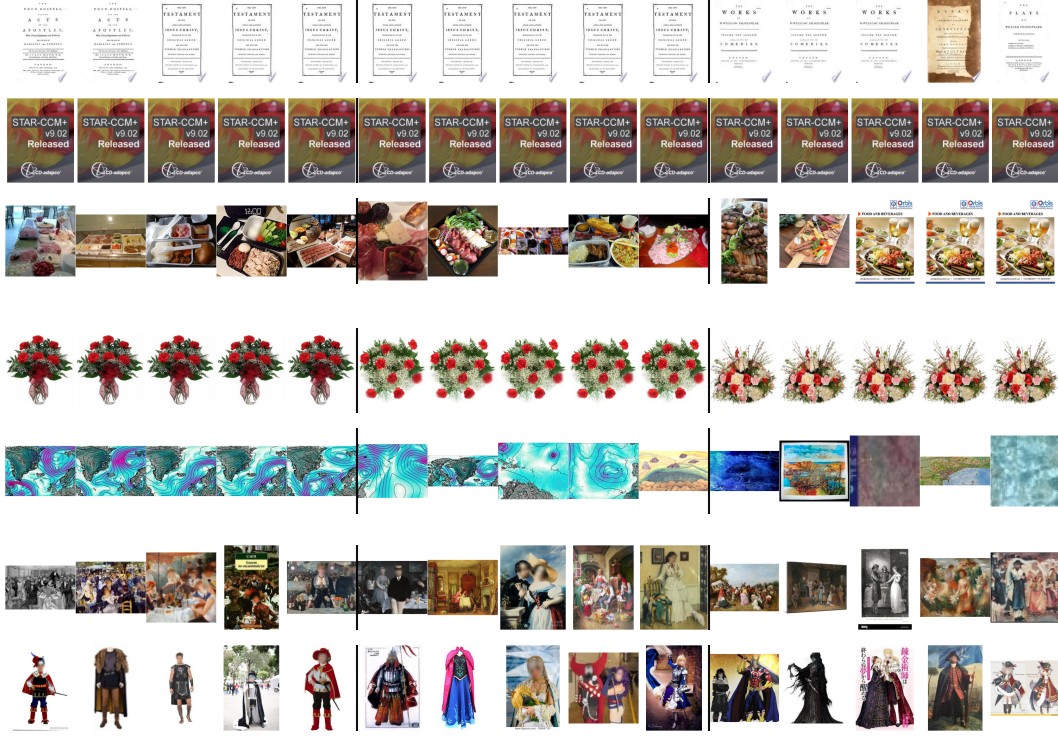

Figure 14: Visualization of different $k$-SAE features. For each row, we display the top 5, middle 5, and bottom 5 images with nonzero activations for the neuron corresponding to the row, ordered in descending activation value.

