# OpenReview forum: "Matched Data, Better Models: Target Aligned Data Filtering with Sparse Autoencoders"
_ICLR.cc/2026/Conference — ICLR 2026 Poster_

### Official Review · Reviewer_aMFk · 2025-10-30

**Soundness:** 2
**Presentation:** 3
**Contribution:** 2
**Rating:** 4
**Confidence:** 3

**Summary:**

This paper tackles the problem of filtering large-scale image-text datasets (like those scraped from the web) to improve model training. Such datasets are often noisy (containing irrelevant or low-quality data) and redundant (many very similar examples), which can hurt model performance. Traditional filtering methods usually score each data sample for quality and drop the low-scoring ones, but fail to capture the 'high-level feature. This paper proposed SDM to capture the high-level features and estimate the target distribution from ImageNet as a standard. Follow the standard, SDM can select high-quality samples from new datasets that follow the standard distribution at a high conceptual level.

**Strengths:**

1. This paper proposed SDM to capture high-level interactions (features), which go beyond single-sample selection.
2. SDM uses high-quality datasets as standard to estimate the target distribution of high-level features
3. This data selection strategy is verified in rich downstream tasks.

**Weaknesses:**

1. It highly relies on the high-quality dataset, the high-level features learned from it are treated as the target distribution.
2. The interpretability of the high-level features is a big concern as I listed in the questions

**Questions:**

1.  In line 154, why the sparse dimension $d_{sparse} >> d_{in}$. For the learned sparse features, how can we verify what the features are? Why are they informative to represent the information shown in the Image or text?
2. For estimating the target distribution of 'high-level' features, this paper uses ImageNet as the standard. If all datasets share the same 'high-level' features?
3. Although ImageNet is comprehensive in images, different datasets may have different features. This paper uses the same sparse autoencoder to align the learned high-level features. What if there are new important features in new datasets?
4. Moreover, the proportion of different classes of images may affect the target distribution. If the gained performance is because the autoencoder is pre-trained on ImageNet, the data selection process uses ImageNet as a standard, which makes the selected data fit the model.
5. For the comparison of different model sizes, does the best baseline have the same results across the different sizes?

---

> ### Author Response · Authors · 2025-11-12
> **Rebuttal**
>
> We thank the reviewer for taking the time to review our paper! We answer all of the questions below to the best of our ability:
>
> ---
>
> *“Why is $d_{sparse} \gg d_{in}$?”*
>
> Standard neural representations (with dimensionality $d_{in}$) are designed to be compact and pack as much information about the image as possible into a dense representation. By design, this means that individual neurons will activate for highly unrelated concepts as shown in prior work and in Figure 3 (left). By contrast, using a much larger $d_{sparse}$ provides the model with sufficient capacity to allocate each neuron to represent individual disentangled concepts.
>
> ---
>
> *“How can we verify what the features are?”*
>
> Past work has demonstrated that SAE features tend to be far more interpretable [1, 2, 3]. This is verified by (1) isolating a random SAE neuron, (2) finding a set of images that highly activate that neuron, and (3) qualitatively examining the set of images or quantifying their homogeneity. This analysis is done in past work [1, 2, 3] and in our work — see Figure 3 (right), Appendix K for qualitative results, and Table 2 for quantitative results.
>
> ---
>
> *“Why are they informative to represent the information in the image?”*
>
> The sparse representations contain roughly the same amount of information about the image as the CLIP representations since they are trained to reconstruct them (and achieve low reconstruction loss). However, the sparse representations are more amenable to being used with feature-based submodular functions, as discussed in Sections 2.1 and 2.3.
>
> ---
>
> *“If all datasets share the same high-level features?”*
>
> We apologize, but we do not understand this question, as we did not use the term “high-level features” in the paper. Could the reviewer please elaborate on what they mean by “high-level features”?
>
> ---
>
> *“What if there are new important features in new datasets?”*
>
> This is a valid concern, as downstream domains may indeed contain concepts that are absent or underrepresented in ImageNet. However, in our case, using ImageNet as the target distribution yields significant performance gains across 38 datasets, suggesting that this limitation does not manifest in our setting.
>
> ---
>
> *“Gains are due to the autoencoder being trained on ImageNet.”*
>
> The SAE is **not** trained on ImageNet — it is trained on the unfiltered DataComp-Medium pool, as stated in lines 315–316.
>
> ---
>
> *“For different model sizes, does the best baseline have the same results across different sizes?”*
>
> We never compare different model sizes in the paper. If the reviewer is referring to Figure 4, where we compare different **dataset sizes**, we use the best baseline with the best dataset size as reported in their paper [4] — thus it is an upper bound on their performance at other sizes. We will update the paper to clarify this. If the reviewer is referring to Figure 5, we emphasize that we do not train different models; we simply change the model that is used to generate the embeddings/scores for data selection.
>
> ---
>
> We hope our responses have fully addressed the reviewer’s concerns and clarified the key contributions of our work. We kindly invite the reviewer to reconsider their evaluation in light of these clarifications.
>
> ---
>
> **References**
> [1] *Scaling and Evaluating Sparse Autoencoders.* ICLR 2025
> [2] *Sparse Autoencoders Find Highly Interpretable Features in Language Models.* ICLR 2024
> [3] *Sparse Autoencoders Learn Monosemantic Features in Vision-Language Models.* arXiv 2025
> [4] *CLIPLoss and Norm-Based Data Selection Methods for Multimodal Contrastive Learning.* NeurIPS 2024

---

> > ### Comment · Reviewer_aMFk · 2025-11-27
> >
> > Thanks for your response. Regarding the question “Do all datasets share the same high-level features?”, by “high-level features” I actually mean the sparse features in the high-level interactions learned by the SAE. My question is: after each image is encoded into a sparse vector $d_{sparse}$, how can we isolate or identify individual concepts from this embedding?
> >
> > In my understanding, $d_{sparse}$ should be shared across samples and function like a dictionary; otherwise, it would not be possible to use ImageNet as a standard for matching and selection. If this is the case, my concern is that $d_{sparse}$ cannot capture genuinely new concepts, since it is fixed and learned during pre-training, which may limit the model’s ability to generalize.
> >
> > Moreover, regarding the example shown in Figure 3, does it really make sense to interpret each neuron as representing a single concept? I am not entirely sure about this assumption.
> >
> > Regarding the statement “Gains are due to the autoencoder being trained on ImageNet,” what I mean is that, during data selection, the distribution of ImageNet is used as the reference (or standard) for matching. Therefore, the observed gains may essentially be driven by aligning to the ImageNet distribution, rather than purely by the autoencoder itself.

---

> > > ### Author Response · Authors · 2025-11-28
> > > **Response**
> > >
> > > Response to aMFk
> > >
> > > We thank the reviewer for clarifying their initial questions. We answer all of them below:
> > >
> > > *“How can we isolate or identify individual concepts from this embedding?”*
> > >
> > > We can identify concepts by performing the analysis shown in Figure 3 and Appendix L, where our findings—consistent with prior work [1-3]—show that individual neurons often correspond to human-interpretable concepts.**However, isolating these individual concepts and mapping every neuron is not necessary for our approach.** Instead, our analysis provides a plausible explanation for why SAE features are effective for data selection. Moreover, in response to reviewer hCWT ’s request, we include empirical evidence in Section 3.5 in the updated version demonstrating that replacing SAE features with dense CLIP embeddings leads to a substantial drop in performance.
> > >
> > >
> > > *”Sparse features cannot learn new features*”
> > >
> > > The reviewer is absolutely correct - it is very likely that the dataset that is used to train the SAE will restrict the concepts that the SAE will learn. For this reason, we use the **pretraining dataset** i.e DataComp-medium to train the SAE. It is reasonable to assume that the set of concepts contained in DataComp-medium is a superset of the set of concepts contained in downstream tasks - if this were not true any resulting CLIP model would exhibit poor zero-shot performance. Therefore, our framework first learns features based on DataComp and then uses ImageNet only to identify which of the features are important.
> > >
> > >
> > > *”Gains are due to matching the distribution to ImageNet”*
> > >
> > > Using a reference dataset is a very common practice in data selection. In the context of DataComp, ImageNet is used as the reference dataset by several prior approaches, including NormSim [4], FLYT [5], and the best-performing method introduced in the original DataComp paper [6]. Reference datasets are also standard in other data selection settings, such as (1) selecting data for LLM pretraining [7,8] and SFT [9], and (2) retrieval-based augmentation for CLIP adaptation [10,11]. Similar assumptions also appear in related areas such as transductive learning. **Importantly, our gains are not solely attributable to the use of a reference dataset, since many of our baselines rely on the exact same reference dataset yet achieve substantially lower performance (see Table 1 and Figure 5)**. Our paper introduces a novel method of using the reference dataset based on SAE features.
> > >
> > >
> > > We sincerely hope that we have sufficiently answered all of your concerns, and kindly ask the reviewer to consider raising their score to reflect this.
> > >
> > > ## References
> > > - [1] Scaling and Evaluating Sparse Autoencoders. ICLR 2025
> > > - [2] Sparse Autoencoders Find Highly Interpretable Features in Language Models. ICLR 2024
> > > - [3] Sparse Autoencoders Learn Monosemantic Features in Vision-Language Models. arXiv 2025
> > > - [4] CLIPLoss and Norm-Based Data Selection Methods for Multimodal Contrastive Learning. NeurIPS 2024
> > > - [5] Filter Like You Test: Data-Driven Data Filtering for CLIP Pretraining, arXiv 2025
> > > - [6] DataComp: In search of the next generation of multimodal dataset, NeurIPS 2023
> > > - [7] DataComp-LM: In search of the next generation of training sets for language models, NeurIPS 2024
> > > - [8] Language Models Improve When Pretraining Data Matches Target Tasks, arXiv 2025
> > > - [9] LESS: Selecting Influential Data for Targeted Instruction Tuning, ICML 2024
> > > - [10] COmBinatorial Retrieval Augmentation for Few-Shot Adaptation, CVPR 2025
> > > - [11] Neural Priming for Sample-Efficient Adaptation, NeurIPS 2023

---

### Official Review · Reviewer_7pgs · 2025-10-31

**Soundness:** 3
**Presentation:** 3
**Contribution:** 3
**Rating:** 6
**Confidence:** 3

**Summary:**

This paper addresses data filtering for vision-language models pretrained on large, noisy datasets. The authors propose Submodular Distribution Matching (SDM), a submodular framework that selects samples by matching feature distributions learned via sparse autoencoders. Experiments on DataComp-medium demonstrate that SDM achieves state-of-the-art or near state-of-the-art accuracy across ImageNet-1K and 38 downstream tasks, while significantly reducing computational cost.

**Strengths:**

1. The proposed Submodular Distribution Matching (SDM) presents a novel and promising approach to data filtering. Extensive experiments and superior performance compared with baseline methods convincingly demonstrate its effectiveness.

2. The theoretical analysis linking the designed submodular maximization objective to the distribution-matching target strengthens the rationale behind the proposed method.

3. The paper is well organized and clearly written, making it easy to follow and understand.

**Weaknesses:**

From a model training perspective, selecting a data subset that precisely matches or aligns with the target distribution may critically influence the model’s out-of-distribution (OOD) generalization capability. Providing additional empirical evaluation of OOD performance using the filtered data would help clarify the practical impact of the proposed method and further highlight its contribution to improving generalization beyond the training distribution.

**Questions:**

1. My main concern is that selecting a data subset that exactly matches or aligns with the target distribution may critically affect the model’s out-of-distribution generalization. How to balance the goal of distribution matching with maintaining data diversity during filtering?

---

> ### Author Response · Authors · 2025-11-12
> **Rebuttal**
>
> We thank the reviewer for their thoughtful assessment of our paper!
>
> Regarding the concern about out-of-distribution (OOD) generalization and data diversity, we emphasize that our evaluation already spans **38 diverse downstream datasets**, providing strong evidence that the proposed framework does not overfit to a single distribution. Table 1 (“IN1K Shifts”) specifically evaluates OOD performance, where SDM achieves substantial gains over the best baseline. Furthermore, when considering all downstream datasets excluding ImageNet, SDM attains 36.4% compared to 35.8% for the best baseline.
>
> We sincerely appreciate the reviewer’s recognition of our method’s novelty, theoretical grounding, and extensive experimental validation, and we hope this clarification supports a higher evaluation.

---

### Official Review · Reviewer_hCWT · 2025-11-01

**Soundness:** 2
**Presentation:** 3
**Contribution:** 2
**Rating:** 6
**Confidence:** 3

**Summary:**

The paper proposes SDM (Submodular Distribution Matching), a data filtering framework for vision-language models that combines sparse autoencoders (SAEs) with submodular optimization. The method learns disentangled and monotone features through SAEs with a novel monotonicity loss, then selects data subsets that match a target distribution while considering sample quality. The authors claim state-of-the-art results on DataComp-medium benchmark.

**Strengths:**

1. Novel framework integration: First to combine SAEs with submodular optimization for data selection, providing both interpretability and theoretical guarantees
2. Theoretical contribution: Establishes connection between KL divergence minimization and submodular maximization (Theorem 2.3), enabling efficient algorithms
3. Practical impact: Achieves competitive performance on DataComp-medium with reasonable computational budget compared to alternatives
4. Comprehensive evaluation: Tests across 38 downstream tasks, showing consistent (if modest) improvements
5. Monotonicity loss innovation: Novel loss term (Eq. 3) for encouraging monotone features in SAEs could be valuable for interpretability community

**Weaknesses:**

Major
1 Mathematical Soundness:
- The objective \log m_i(A) is undefined when mi​(A)=0. Add explicit ε-smoothing: log(mi​(A)+ϵ) with sensitivity analysis.
- Unverified bound: The proof of Lemma 2.4 relies on∥h∥∞​≤β which the SAE architecture doesn't guarantee.
2. Statistical Validation: All results lack error bars. Re-run with ≥3 seeds, report mean±std for all tables, and provide significance tests.
3. Computational reporting: Should clarify total pipeline costs including encoding time


Minor

1. No ablation separating component contributions
2. Provide more details on Algorithm 1 (distance metrics, buffer size)
3. Explain the 5-run intersection choice

**Questions:**

1. How sensitive are results to ε-smoothing value? Please provide ablation.
2. Can you guarantee the β bound in practice? What's the actual max activation value observed?
3. Why take intersection of 5 greedy runs rather than union or single run?
4. What's the breakdown of improvements from SAE features vs submodular selection?
5. How does performance vary with different target distributions?

This paper makes a solid contribution to data selection for large-scale training. The idea of using SAEs to obtain interpretable features for submodular selection is clever and well-executed. While there are technical details to clarify, the core contribution is valuable and the experimental results support the claims.

The 0.7% average improvement may seem modest, but in the context of large-scale training where compute costs are substantial, even small improvements are valuable. The framework is also general and could be applied to other domains beyond vision-language models.

---

> ### Author Response · Authors · 2025-11-22
> **Rebuttal (1/2)**
>
> We thank the reviewer for the highly detailed and overall positive review! Below we answer all of the key concerns that were raised.
>
> *“Log is undefined without epsilon smoothing.”*
>
> In our work, we always use $\epsilon = 1$, which ensures that the $\log$ function’s output is always non-negative. In addition, any value of $\epsilon \neq 1$ would make the function non-normalized; that is, for an empty set, it would have a value that is not equal to $0$. For submodular maximization, the function must be monotone, non-decreasing, and normalized (zero empty-set evaluation) to have the $1 - 1/e$ guarantee, hence explaining our decision of having $\epsilon = 1$ [10]. We thank the reviewer for pointing out that we had this implementation detail omitted, and we will include this key detail in the next version of the paper.
>
> *“Is assuming upper bound on $h_{\infty}$ realistic”*
>
> Similar to standard neural network training, we apply regularization during the training of our SAEs. In particular, we apply regularization on the norm of the activation [8, 9], making the upper bound realistic. Empirically, the maximum value is $6.88$, which is much less than $k = 128$, as well as the upper bound on the cardinality $|A| = 18\text{M}$. Finally, we observe in Figure 6 (Appendix F) that maximizing the submodular surrogate does indeed result in solutions with very low KL-divergences to the target distribution.
>
> *“No error bars”*
>
> Conducting $2+$ additional trials for each experiment would require over $2{,}000$ A100 GPU hours, which is difficult to support within an academic compute budget and the short discussion timeline. **Following the conventions established in prior large-scale pretraining work related to DataComp [1–7], we report single-run results, which is the standard practice at this scale.**
>
> We did however repeat our main result in Table~1 three times and observed the following means and standard deviations:
>
> |        | IN1K   | IN1K Shifts | VTAB   | Retrieval | Avg    |
> |--------|--------|-------------|--------|-----------|--------|
> | **Mean** | 35.34% | 27.23%      | 38.47% | 26.83%    | 36.23% |
> | **Std**  | 0.12%  | 0.15%       | 0.71%  | 0.15%     | 0.21%  |
>
> We note that the standard deviation is significantly lower than the margin of improvement that SDM attains over the next best baseline.
>
> *“Computational reporting”*
>
> Below, we summarize the total computational budget used in our experiments:
>
> - CLIP ViT-L/14 feature extraction: $60$ A100 hours
> - NegCLIPLoss score computation: $5$ A100 hours
> - SAE training: $15$ A100 hours
> - Submodular maximization: $5$ CPU hours
>
> Importantly, both feature extraction (a shared bottleneck across all methods) and NegCLIPLoss scoring scale linearly with the size of the candidate pool. In contrast, the SAE training cost can remain fully independent of the pool size. This breakdown is scattered throughout Section~3, but we can merge it into a single table for clarity in the next version of the manuscript.
>
> *“Why do an intersection over 5 runs?”*
>
> For each step of the stochastic greedy algorithm, we randomly sample a subset of approximately $35$ candidates (controlled by the $\epsilon$ parameter) from $(V \setminus A)$ and evaluate the gain only on these. Occasionally, we may get unlucky and random sampling will select a suboptimal candidate set, leading to the addition of a bad sample.
>
> To mitigate this, we take the **intersection of multiple stochastic greedy runs**, which effectively filters out such suboptimal selections. Moreover, this procedure is parallelizable across cores, allowing us to perform several runs in parallel with minimal computational overhead.
>
> Below, we show the comparison between a single run and the intersection over $5$ runs:
>
> | Method                   | IN1K (%) | Avg. (%) |
> |--------------------------|----------|----------|
> | Single run               | 33.5     | 36.0     |
> | Intersection over 5 runs | **35.2**     | **36.4**     |
>
> We noticed that this explanation and ablation was missing from the paper, but we will add the explanation and table to the supplementary in the next version.

---

> > ### Author Response · Authors · 2025-11-22
> > **Rebuttal 2/2**
> >
> > *“More details on Algorithm 1”*
> >
> > We used a buffer size of $50\text{K}$ samples and the set sizes for $E$ and $M$ were both set to $128$.
> >
> > *“Component-wise Analysis”*
> >
> > The reviewer requested an analysis that isolates the contributions of (i) using SAE-derived sparse features and (ii) using submodular selection.
> >
> > To remove SAE features, we instantiate our feature-based objective directly on **dense CLIP embeddings**.
> > To remove submodularity, we simply remove the $\log$ function in the feature-based objective (eliminating diminishing returns).
> >
> > We report results on **ImageNet-1K** and the **Average over 38 datasets** below.
> >
> > ### ImageNet-1K
> >
> > |               | **Submodular** | **No Submodular** |
> > |---------------|----------------|--------------------|
> > | **Sparse**    | **35.2**       | 34.6               |
> > | **Dense**     | 24.1           | 23.7               |
> >
> > ### Average Over 38 Datasets
> >
> > |               | **Submodular** | **No Submodular** |
> > |---------------|----------------|--------------------|
> > | **Sparse**    | **36.4**       | 35.1               |
> > | **Dense**     | 29.1           | 28.9               |
> >
> > Overall, we observe that **sparse autoencoders are essential**, yielding an improvement of approximately $11\%$ in average accuracy. We also see that **submodularity provides significant gains**, especially when using sparse features. This is a very interesting set of experiments, so we thank the reviewer for suggesting it! We will include it in the next version of the paper.
> >
> > *“How does performance vary with target distribution?”*
> >
> > The choice of target distribution plays a central role in the effectiveness of SDM. Following prior work [1, 5, 6], our main experiments use ImageNet as the target distribution for subset selection.
> >
> > However, we also tested a broader target distribution constructed by concatenating the training splits of **24 downstream datasets** (including ImageNet), as done in [1]. Interestingly, this reduced overall performance:
> >
> > | Target             | IN1K | IN1K Shifts | VTAB | Retrieval | Avg  |
> > |--------------------|------|-------------|------|-----------|------|
> > | ImageNet-1K        | 35.2 | 27.1 | 38.6 | 26.8 | 36.4 |
> > | 24 datasets        | 33.3 | 26.7 | 37.1 | 26.7 | 35.8 |
> >
> > We hypothesize that this decrease stems from:
> > - severe imbalance when merging datasets of different sizes and concept distributions,
> > - inclusion of low-quality datasets (e.g., CIFAR-10), whose samples degrade performance on others.
> >
> > Developing more principled strategies for constructing larger target distributions (e.g., via data mixture optimization) remains an interesting direction for future work that we plan to pursue.

---

> > > ### Author Response · Authors · 2025-11-22
> > > **Rebuttal References**
> > >
> > > ## References
> > >
> > > [1] *Cliploss and norm-based data selection methods for multimodal contrastive learning*, NeurIPS 2024
> > > [2] *Data Filtering Networks*, NeurIPS 2023
> > > [3] *T-MARS: Improving Visual Representations By Circumventing Text Feature Learning*, ICLR 2024
> > > [4] *HYPE: Hyperbolic Entailment Filtering for Underspecified Images and Texts*, ECCV 2024
> > > [5] *DataComp: In search of the next generation of multimodal dataset*, NeurIPS 2023
> > > [6] *Filter Like You Test: Data-Driven Data Filtering for CLIP Pretraining*, arXiv 2025
> > > [7] *Optimizing ML Training with Metagradient Descent*, arXiv 2025
> > > [8] *Revisiting Activation Regularization for Language RNNs*, ICML 2017
> > > [9] *Deep Submodular Peripteral Networks*, NeurIPS 2024
> > > [10] *An Analysis of Approximations for Maximizing Submodular Set Functions*, Mathematical Programming 1978

---

### Author Response · Authors · 2025-11-25
**Summary of Changes**

Dear Reviewers and AC,

We have carefully addressed all concerns raised during the review process and incorporated the corresponding changes into the revised manuscript. In particular, we have made the following updates:
- **Section 2.2:** Added previously omitted discussion on the activity regularizer, including its practical role in reducing the effective value of $\beta$ (with further clarification in Section 2.4).
- **Section 2.4:** Clarified that our implementation uses $\log(1 + m_i(A))$ rather than $\log(m_i(A))$, ensuring nonnegativity and normalization ($f(\emptyset)=0$).
- **Section 3.1:** Consolidated all computational cost analyses into a single paragraph to improve readability.
- **Section 3.5:** Added a component-wise ablation to more clearly isolate the contributions of sparse features and submodularity.
- **Appendix K:** Added ablation and explanation of the intersection over 5 runs, as well as additional experiments with alternative target distributions.
- **Appendix G** Added all of the specific hyperparameters that were used for SAE training.

All substantive revisions are highlighted in blue in the updated manuscript. If there is anything we may have overlooked, we kindly invite the reviewers to let us know before the discussion period concludes.

We sincerely appreciate the reviewers’ thoughtful feedback and the time invested in evaluating our work.

---

### Meta-Review · Area_Chair_Ct34 · 2026-01-09

**Summary:**

This paper addresses the fundamental problem of data filtering for vision-language model pretraining on large, noisy datasets. The method demonstrates notable innovation by combining sparse autoencoders with submodular optimization for data selection. Considering the strength of the review and inspiration, the AC recommends acceptance.

**Reviewer Concerns:**

Addressed Concerns:
- The objective \log m_i(A) is undefined when mi​(A)=0. Add explicit ε-smoothing: log(mi​(A)+ϵ) with sensitivity analysis.
- Unverified bound: The proof of Lemma 2.4 relies on∥h∥∞​≤β which the SAE architecture doesn't guarantee.
- Statistical Validation: All results lack error bars. Re-run with ≥3 seeds, report mean±std for all tables, and provide significance tests.
- Computational reporting: Should clarify total pipeline costs including encoding time
- No ablation separating component contributions
- Provide more details on Algorithm 1 (distance metrics, buffer size)
- Explain the 5-run intersection choice
- Regarding the concern about out-of-distribution (OOD) generalization and data diversity
- How to isolate or identify individual concepts from this embedding
- Sparse features cannot learn new features
- Gains are due to matching the distribution to ImageNet

**Reviewer Scores:**

The paper received initial scores of 6, 6, 4. The concerns from the two 6-score reviewers were comprehensively addressed in the rebuttal.

The reviewer who assigned a score of 4 raised more substantial concerns, including: (1) the methodology for identifying individual concepts from the embedding space, (2) the sparse features update mechanism, and (3) the source of the reported performance gains. The authors provided detailed responses to each of these points with additional explanations and clarifications. After carefully reading through the reviewer-author discussion, I find that these concerns have been reasonably addressed, and the explanations provided are convincing.

---

### Decision · Program_Chairs · 2026-01-26

Accept (Poster)